artificial intelligence/computer modelling and simulation/systems theory

deception, disinformation, public goods games, microeconomics, multi-agent systems, tragedy of the digital commons

**Authors for correspondence:**
Ştefan Sarkadi
e-mail: stefan.sarkadi@inria.fr
Iyad Rahwan
e-mail: rahwan@mpib-berlin.mpg.de

# The evolution of deception

Ştefan Sarkadi[1,2,3], Alex Rutherford[2,3], Peter McBurney[2], Simon Parsons[2,4] and Iyad Rahwan[3,5]

[1]Inria, Sophia-Antipolis, France
[2]Department of Informatics, King's College London, London, UK
[3]Media Lab, MIT, Cambridge, MA, USA
[4]School of Computer Science, University of Lincoln, Lincoln, UK
[5]Center for Humans and Machines, Max Planck Institute for Human Development, Berlin, Germany

ŞS, 0000-0003-3999-528X

Deception plays a critical role in the dissemination of information, and has important consequences on the functioning of cultural, market-based and democratic institutions. Deception has been widely studied within the fields of philosophy, psychology, economics and political science. Yet, we still lack an understanding of how deception emerges in a society under competitive (evolutionary) pressures. This paper begins to fill this gap by bridging evolutionary models of social good—*public goods games* (PGGs)—with ideas from *interpersonal deception theory* (Buller and Burgoon 1996 *Commun. Theory* **6**, 203–242. (doi:10.1111/j.1468-2885.1996.tb00127.x)) and *truth-default theory* (Levine 2014 *J. Lang. Soc. Psychol.* **33**, 378–392. (doi:10.1177/0261927X14535916); Levine 2019 *Duped: truth-default theory and the social science of lying and deception*. University of Alabama Press). This provides a well-founded analysis of the growth of deception in societies and the effectiveness of several approaches to reducing deception. Assuming that knowledge is a public good, we use extensive simulation studies to explore (i) how deception impacts the sharing and dissemination of knowledge in societies over time, (ii) how different types of knowledge sharing societies are affected by deception and (iii) what type of policing and regulation is needed to reduce the negative effects of deception in knowledge sharing. Our results indicate that cooperation in knowledge sharing can be re-established in systems by introducing institutions that investigate and regulate both defection and deception using a decentralized case-by-case strategy. This provides evidence for the adoption of methods for reducing the use of deception in the world around us in order to avoid a *Tragedy of the Digital Commons* (Greco and Floridi 2004 *Ethics Inf. Technol.* **6**, 73–81. (doi:10.1007/s10676-004-2895-2)).

# 1. Introduction

Deception plays a critical role in information dissemination. Deception also plays a crucial role in survival and continuation of species. Significant studies from evolutionary biology have even focused on the bio-physiological properties of deception in plants and animals [1] that determine certain types of behaviours. Humans, on the other hand, employ deceptive behaviour at a higher level, which is not necessarily determined or highly dependent on their bio-physiological properties.[1] This is especially the case in social, political and economic contexts where deception takes the form of knowledge manipulation. Areas such as Philosophy [2,3], Psychology [4,5], Communication Theory [6–9], Economics [10,11], Information Theory [12] and Security Studies [13–15] have looked at the these higher-level properties and components of deception.

Information technologies such as smartphones, social media and rolling news coverage have greatly increased the access to information and informed decision making which in turn determine opinion formation and behavioural change. However, this increased access also offers more opportunities for the knowledge public good to be compromised by deception. Additionally, due to this advancement in technology, other kinds of actors, apart from humans, that share and generate knowledge have come into existence. In particular, artificial intelligence (AI) has gained a strong momentum in the past decades, and this has caused the emergence of intelligent artificial and autonomous agents. The risks related to the sharing of knowledge and information as a social good posed by AI is that artificial autonomous agents might develop their own reasons to act deceptively as it is pointed out in [16] and, more recently, in [17,18]. AI has also seen an emerging interest in the problem of *fake news* and the potential ability of machines (i) to be used for fake news generation [19] and fake news detection [20,21]; or even (ii) to use higher-order cognitive mechanisms to manipulate the beliefs of others in order to deceive [22–24]. All of this leaves us with an urgent need to understand deception better, and to devise methods for reducing its impact.

To understand how machines or artificial agents might deceive *in the wild*, we adopt the perspective of machine behaviour [25]. Machine behaviour takes a rather different approach to mainstream work on AI. Instead of aiming to maximize or optimize algorithmic performance of AI agents, machine behaviour focuses on defining measures of micro and macro outcomes to answer broad questions such as how these agents behave in different environments and whether human interactions with agents alter societal outcomes. Therefore, based on the results of our modelling approach, we aim to interpret the actions of intelligent agents as part of a wider ecosystem that ranges from the technical aspects that underlie the design of AI agents to the legal, institutional and societal constraints that govern interactions between humans and machines.

We know that a shared system of accurate and non-partisan knowledge confers advantages for all members of society and promotes informed and democratic decision making. It is a crucial component of the openness and transparency that enables the self-governance of such societies [26]. However, there exists a unilateral incentive for an individual to provide misinformation that confers advantages to that individual. This misinformation will damage the transparency and openness needed for self-governance. Therefore, mechanisms are required to promote cooperation in information dissemination and to mitigate the negative effects of misinformation and deception. One such mechanism for countering the negative effects of deceptive behaviour is that of punishing such behaviour, and it is this mechanism that we will focus on in this paper. However, it is very important to note that other ways to counter deception might be more appropriate in different contexts, but their exploration is beyond the scope of our paper.

At this point in time, it has become increasingly important to address the future impact that deceptive social agents (human or artificial) can have on society in general. Not doing so risks endangering our societies' ability to self-govern. To mitigate this problem, we must ask ourselves what types of rules need to be implemented or what types of mechanisms should be employed in order to reduce the impact of deceptive agents on knowledge sharing and generation. In this paper, we begin to provide some answers to these questions. We do this, by studying a *public goods game* (PGG) model based on the those in [27,28]. We use these games to explore how the evolution of cooperation in different populations of agents is influenced by deception. A public good commonly models some type of financial or physical good that is shared by the members of a society, but can also represent a shared system of knowledge within a society, and that is how we use it here. This model allows us to answer

---

[1]This does not imply that deception in humans is not driven by some bio-physiological properties.

the following questions: (i) Does deception lead to the breakdown of cooperation in societies where regulatory institutions exist? and (ii) Can cooperation be reached in societies where deception is present? Note that, in common with [29,30], we are less concerned with identifying the optimum way to organize a society than to identify mechanisms that can be put in place to ensure that the society can continue to function, in our case, in the face of deception.

# 2. Background

We consider deception to be a strategic and social behaviour that agents (natural or artificial) employ in order to gain advantage over the members they interact with. In game-theoretic terms, deception is considered by the literature to be a non-cooperative behaviour. However, agents that intend to deceive, usually try to emulate cooperative behaviour. To the target agent that observes it, a deceptive agent appears cooperative, while, in reality, it is non-cooperative. Therefore, the type of agent-based model we chose for this study has to be able to represent deception as a falsely cooperative behaviour while allowing us to test under which circumstances real cooperation is promoted.

The major studies on cooperation focus on the aspects of complexity in agent-based systems and treat cooperation as an emergent behaviour in such multi-agent systems [28,31,32]. Thus, even though the overall behaviour of a population of agents can be regarded as complex, it emerges from a set of strategic behaviours that are intrinsically simplistic. The literature shows that PGGs as evolutionary game theoretical models have been successfully used to study how cooperation emerges, and that they are a powerful framework to understand the conditions under which cooperation is stable through mechanism design. This is why we adopt a PGG approach here.

The set of behaviours that most work on PGGs has focused on are *cooperation*, *defection* (also known as *free-riding*) and *punishment*. Different variants of punishment have been studied, including *pool-punishment*, where individuals pay a tax to maintain third parties (the punishers) who carry out punishment, and *peer-punishment*, where individuals punish their peers. However, if we look at how agents in human societies behave, we can identify other types of behaviours that are not as straightforward, such as deception. Apart from simply cooperating, defecting and punishing, humans are also able to mask their intentions behind these behaviours. For instance, an agent can pretend to cooperate while defecting in a PGG. Thus, to other participating players the deceiver seems to be one of the cooperators, thus enjoying both the social benefits of a cooperator and the financial benefits of a defector. As a cooperator, the deceiver might be regarded as being ethical and pro-social, while as a defector the deceiver receives a certain political or financial satisfaction.[2] In terms of knowledge sharing, a cooperator is transparent and fair, thus it shares truthful information with the other members of society. Contrastingly, a deceiver will contribute with untruthful information when engaged in knowledge sharing.

Given that deception is intrinsically a communicative behaviour, we refer to the literature in communication theory to see what factors must be included in our PGG model. We mainly consider factors that directly influence deceptive behaviour in social interactions. One such cognitive factor, according to truth-default theory, is the default attitude of trust [7]. This truth-default attitude of individuals means that people are usually not actively engaged in deception detection, therefore in everyday communicative interactions, they expect the interlocutor to be honest. Apart from a bias towards trustworthiness, there are other socio-cognitve factors such as *cognitive load*,[3] *leakage*[4] and *communicative skill*[5] that have been identified by interpersonal deception theory (IDT) in [6].

# 3. Research questions

The two main research questions that we aim to answer in this study are: **RQ1** *Does deception lead to the breakdown of cooperation in societies where regulatory institutions exist?* and **RQ2** *Can cooperation be reached in societies where deception is present?*

---

[2]According to [11], deception is a selfish behaviour that aims to maximize one's payoff.

[3]The cognitive effort that is spent in order to solve a task, such as forming or planning a deceptive strategy.

[4]The information leaked by a deceptive agent due to cognitive load.

[5]The social skill of an agent to form, plan and deliver messages. According to IDT, the communicative skill regulates cognitive load and reduces leakage.

To answer RQ1, we have reproduced the voluntary PGG without second-order pool-punishment introduced in [28] and used by [27] to study the emergence of corruption in centralized versus decentralized systems. The reproduction of this PGG shows us how cooperation can emerge in societies with different types of regulation mechanisms for free-riding. Regarding this PGG, we have formulated the question: *What would happen if the strategy of deception is introduced in the system for agents to adopt?* and we formulated and introduced the strategy of deception. We aimed to answer questions that directly follow from the introduction of deception in this PGG, namely: would the introduction of deception have any influence at all? Are the classical punishment mechanisms robust enough to deal with both defection and deception, or would they break down and let free-riders dominate cooperation in the long term? Would deception become the evolutionary stable strategy and under which conditions?

The breakdown of cooperation and emergence of free-riding in governed societies, whether these are physical, digital or hybrid, would have negative consequences in the long run. One of these consequences in the case of digital and hybrid societies is the *Tragedy of the Digital Commons* (TDC) [33]. Hence, we aimed to answer RQ2 by formulating and independently introducing different strategies that aim to counter deception in the PGGs where deception is present. More specifically, the introduction of these strategies meant to answer the following question *What would happen if different strategies for identifying and/or punishing deceivers would be introduced in the system for agents to adopt?* Remember that deception is not the only free-riding strategy in this type of PGG. Defectors are still present. Questions that directly follow from this issue are: What combination of punishment and deception detection (interrogation) strategies is better at countering free-riding? Is it better to keep punishment strategies separate from interrogation strategies, or do these need to be combined to be better at countering free-riding? Do some type of free-riding countering strategies outperform others, such as is the case in [27] where decentralized regulatory systems outperform centralized ones? and Is the introduction of strategies sufficient for countering free-riding or are there other evolutionary factors that affect this, such as social learning as is the case in [28]?

To answer the questions that follow from our two main questions, we have followed the methodology described in the next section.

# 4. Methods

## 4.1. Agent-based modelling of public goods games

We define and run extensive simulations of agent-based models of PGGs by applying mechanism design and game theory to study the evolution of behaviour of populations of agents of a fixed size $N$ in six different PGGs, each with a different set of strategies. Our method can be considered a *content-based* approach to study the evolution of social systems [34]. We chose to follow this type of approach due to the fact that deception and deception detection implied the modelling of complex cognitive aspects of agent behaviour. As noted by the authors in [34], cognitive aspects increase the complexity of the model and make a *value-based* approach intractable. Ideally, in situations where cognitive aspects are not modelled, both content-based and value-based approaches are desirable to study the evolution of agent behaviour.

In a PGG, each participant is faced with two options: (i) contribute to the public pool a given amount $c > 0$; or (ii) not contribute to the public pool. After the participant picks an option, it receives an amount $r \times c \times M_C / M$, where $r$ represents a multiplier representing the increasing returns of cooperative behaviour, $M_C$ represents the number of contributors and $M$ the total number of participants. If $M_C = M$ it means that the social good is maximized and each participant receives the amount equal to $r \times c$. Whatever the case, each participant receives an equal share $r \times c \times M_C / M$ regardless of whether they contributed to the public pool. In the absence of punishment, free-riding (taking the payoff without contributing to it) becomes the dominant strategy.

For each PGG, we perform explicit computations of what payoffs the agents will receive given a subpopulation that is selected to play the game at each iteration. The relative differences between the payoffs obtained by the agents with different strategies in the subpopulation determine the probability that an agent will adopt a different strategy given a function of the *imitation strength* $s \geq 0$ which represents social learning, together with the *exploration rate* $\mu \geq 0$ which represents the natural inclination of agents to randomly adopt another strategy. The imitation strength, or social learning, can be either weak/intermediate or strong. For weak/intermediate social learning the value of $s$ is a

fixed number, while for strong social learning the value of $s$ approaches $\infty$. Social learning represents the tendency of an agent that is selected for mutation to adopt a strategy that compared with its current strategy maximizes the agent's payoff. The higher the value for $s$, the stronger the tendency of adopting the better strategy. When $s \xrightarrow{\infty}$ the agent will always adopt the better strategy. The exploration rate can be viewed as a mutation which models random mistakes in actions as well as purposeful exploration regardless of relative payoffs. This stochastic approach allows us to dynamically represent how the frequencies of the different types of agents evolve over time. The components of our PGGs are the following: a non-empty set of strategies $S \neq \emptyset$; $N$ that is the number of agents in a population to play a PGG; $n_{S_i}$ represents the number of agents in a population with a given strategy $S_i$; $M$ the number of agents that is selected to play a PGG from a population $N$; $r$ is a multiplication factor that is always $1 < r < M - 1$; $c$ represents the investment a cooperative agent contributes to a PGG; $c_{S_i}$ denotes the cost of a given strategy $S_i$; $\Pi_{S_i}$ represents the payoff of a given strategy $S_i$; $s$ represents the imitation strength, which in our model represents social learning; $\mu$ is the mutation rate at which an agent is selected to learn the strategy of another agent; $B$ is the pool-punishment for defection; $b$ is the peer-punishment for defection; $c_b$ is the cost of punishing a defector; $G$ is the cost of pool-punishment; $\Gamma$ is the punishment or tax for deception; and finally $\sigma$ represents the payoff for non-participation. These parameters are summarized in table 1.

We start from Sigmund *et al.*'s voluntary PGG [28] as a baseline before introducing other PGGs, each with different compositions of agents to see how the strategies influence each other. The full set of strategies that we use are as follows, with cooperators, defectors, loners, peer-punishers and pool-punishers being the set studied in [28] and deceivers, interrogators and the two hybrid strategies being novel strategies introduced here. See electronic supplementary material, S1 appendix for additional details.

— **Cooperator (C):** the cooperator receives the PGG payout (equation (4.1), below) and pays the PGG contribution $c$. The cooperator also pays $\beta$, which represents the tax that pays for punishers to exist.
— **Defector (D):** the defector receives the PGG payout, without paying the PGG contribution. However, the defector pays a tax inflicted by the pool-punishers $B$ or the peer-punishers $b$, or by both types of punishers depending on the PGG that is being played.
— **Loner (L)** (a.k.a non-participation): the loner always receives the same payoff $\sigma$, no matter what PGG is being played. The role of the loner is to give a chance to other strategies to invade the population, e.g. to secure neutral drift towards cooperation. This strategy is similarly used in [27,28,35].
— **Pool-punisher (PoP):** the pool-punisher receives the PGG payout, pays the PGG contribution $c$, as well as the cost of pool-punishment $G$. On top of this, the pool-punisher receives a reward that is the tax payed by the cooperators multiplied by the number of cooperators playing the game and divided by the total number of punishers and interrogators selected to play the game, depending on the PGG that is being played.
— **Peer-punisher (PeP):** the peer-punisher receives the PGG payout, pays the PGG contribution $c$, as well as the cost of peer-punishment $c_b$ multiplied by the number of defectors. On top of this, the peer-punisher receives a reward that is the tax payed by the cooperators multiplied by the number of cooperators playing the game and divided by the total number of punishers and interrogators selected to play the game, depending on the PGG that is being played.
— **Deceiver (Dec):** the deceiver receives the PGG payout and does not pay the PGG contribution (similar to what the defector is doing). On top of that, the deceiver is not punished by either type of punisher. Instead, the deceiver can be interrogated by an interrogator and can pay the cost of deception if it is caught. The cost of deception depends on the cognitive load of the deceiver as well as on the risk of leakage from the deceiver. The cost of deception is also influenced by the deceiver's communicative skill.
— **Interrogator (Int):** the interrogator receives the PGG payout and pays the PGG contribution $c$. The interrogator also receives the reward paid by the cooperators and divided by the total number of interrogators and punishers, depending on the PGG that is being played. The interrogator also pays the cost of interrogation, which consists of the cost of interrogating agents in the PGG and the cost of punishing the interrogated agent in the PGG which turn out to be deceptive.
— **Pool-hybrid interrogator ($H_{PoP}$):** this type of interrogator plays both the role of interrogator and the role of pool-punisher. Therefore, it inherits the costs of both types of agents, while receiving the PGG payout and, of course, paying the PGG contribution.
— **Peer-hybrid interrogator ($H_{PeP}$):** this type of interrogator plays both the role of interrogator and the role of peer-punisher. Therefore, it inherits the costs of both types of agents, while receiving the PGG payout and, of course, paying the PGG contribution.

**Table 1.** Parameter values for PGGs.

| description | symbol | value | range |
|---|---|---|---|
| number of agents in a population to play a PGG | $N$ | 100 | |
| number of iterations of a PGG | $T$ | $10^5$ | |
| number of agents selected to play a PGG | $M$ | 5 | $M \leq N$ |
| social learning (imitation strength) | $s$ | 1000 or $\infty$ | $s \geq 0$ |
| exploration rate | $\mu$ | 0.001 | $\mu \geq 0$ |
| PGG contribution | $c$ | 1.0 | $c > 0$ |
| PGG multiplier | $r$ | 3.0 | $1 < r < M - 1$ |
| loner (non-participation) payoff | $\sigma$ | 0.3 | $0 < \sigma < 1$ |
| pool punishment effect | $B$ | 0.7 | $B > 0$, from [27] |
| pool punishment cost | $G$ | 0.7 | $G > 0$, from [27] |
| peer punishment effect | $b$ | 0.7 | $b > 0$, from [27] |
| peer punishment cost | $c_b$ | 0.7 | $c_b > 0$, from [27] |
| tax for punishers to be present | $\beta$ | 0.5 | $\beta > 0$ |
| punishment for deception | $\Gamma$ | 0.8 | $\Gamma > 0$ |
| cost to punish a deceiver | $c_\Gamma$ | 0.5 | $c_\Gamma > 0$ |
| cost to interrogate agents | $c_{interr}$ | 0.5 | $c_{interr} > 0$ |
| communicative skill (for deceivers) | commSkill | 0.5 | $0 < commSkill < 1$ |

For our PGG model, each strategy, except for non-participation, falls into one of the meta-strategies of game theory, namely cooperation and free-riding. The **cooperation** meta-strategy, which requires an agent to make a contribution to the social good, includes cooperation, pool-punishing, peer-punishing, interrogation, pool-hybrid interrogation and peer-hybrid interrogation. The **free-riding** meta-strategy, which requires an agent to not contribute anything to the social good while enjoying the benefits of the social good, includes defection and deception. Thus, the payout resulting from the PGG is:

$$\text{payout} = c \times r \times \frac{N - n_{FR} - n_L - 1}{N - n_L - 1}. \tag{4.1}$$

In equation (4.1) where a PGG is played by a fixed population with $N$ agents, $n_{FR}$ represents the total number of free-riders, and $n_L$ represents the total number of loners (non-participants). This payout is consistent with the previous evolutionary models of PGGs [27,28].

## 4.2. Modelling deception and interrogation

### 4.2.1. Trust in society

We consider trust to be proportional to the number of cooperators in games, but, due to the complexity of the game we are modelling, we need to make the distinction between genuine cooperators, represented by cooperators and interrogators/punishers, and total cooperators which includes deceivers, represented by $N - n_D$. Deceivers are pretending to cooperate, thus they influence the overall trust between members of a population. We have derived this definition of trust based on truth-default theory (TDT) in deception literature [7,8], which states that human agents are biased to trust others by default. We use $t = (N - n_D)/N$ to represent the trust between a population of agents. The higher the levels of trust, the easier it is for deceivers to succeed. According to TDT, human agents are in the truth-default state because they do not perceive evidence that indicates the presence of deception. For instance, if one mostly finds oneself in a context where trust and cooperation are the norm between social agents, then one is more likely to be in this truth-default state. However, one can also be triggered out of the truth-default state if evidence towards deception becomes more prevalent. This means that if we weaken the assumption of trust and cooperation in a society, deception should become more difficult to achieve. Hence more evidence

towards the presence of deception is leaked when deception is attempted. It is reasonable to assume that the fewer cooperators are in a society, the more likely it is for agents to weaken their assumption of the norm of trust and cooperation. Consequently, agents with an investigative and sceptical attitude are more likely to assign deceptive motives to others, which also makes it increasingly difficult for deceivers to actually deceive. Conclusively, the role of trust in the PGGs is to model how strong or weak the norm of trust and cooperation is in a society of agents.

### 4.2.2. Deception model

Deceivers receive the payout of the PGG without making the PGG contribution. They are distinguished from defectors because they are not subject to punishment as they conceal their defection. However, this concealment is costly; it increases with the number of other agents that must be convinced, but decreases with overall trust among the population and the deceivers' innate communicative skill. We consider the following components that contribute to a deceiver's payoff:

  (i) commSkill: communicative skill of the deceiver.
   (a) Reduces the cost of deception.
   (b) The higher the communicative skill, the more likely it is for a deceiver to succeed in deception.
 (ii) $\gamma = 1 - \text{commSkill}$: The deceivers' risk of getting caught
(iii) $\text{cogLoad} = (n_C + n_{Int} + n_{Dec} + n_P) \times (1 - t) \times (1 - \text{commSkill})$: The cognitive load of a deceiver. Where:
   (a) $n_C + n_{Int} + n_{Dec} + n_P$ represents the number of agents that need to be deceived. Here, we also add the number of deceivers, because a deceiver considers other deceivers to be cooperators.[6]
   (b) $(1 - t) \times (1 - \text{commSkill})$ represents the cost to communicate deceptively with another agent.[7]
(iv) $\text{leakage} = n_{Int} \times \gamma \times \Gamma$: Represents the leakage of the deceiver.
   (a) Increases the cost of deception.
   (b) Leakage means that the deceiver leaves a track of evidence that might lead an interrogator to find out about deception.

### 4.2.3. Cost of deception

Let the cost of deception $c_{Dec}$ be a function of cogLoad and leakage, where $c_{Dec} = \text{cogLoad} + \text{leakage}$.

### 4.2.4. Interrogation model

Interrogators receive the same payout as the peer-punishers minus the cost of peer-punishing. They are different from peer-punishers as they do not punish defectors. However, interrogators need to hunt down deceivers and punish them; therefore, they need to pay a cost for interrogation. This cost increases with the number of agents in a population they need to interrogate as well as with the number of deceivers they are likely to reveal and punish. We consider the following components that contribute to an interrogator's payoff:

  (i) $c_\Gamma$: cost of punishing a deceiver. It is multiplied by
   (a) The probability of a deceiver's risk of getting caught $\gamma$, which represents the likelihood of revealing a deceiver. This multiplication represents the risk of a deceiver being caught in a given population.
   (b) The number of deceivers $n_{Dec}$.
 (ii) $c_{interr}$: cost of interrogating an agent. It is multiplied by
   (a) The numbers of agents that need to be interrogated. These are both cooperators and deceivers $n_C + n_{Dec}$.

### 4.2.5. Cost of interrogation

Let the cost of being an interrogator $c_{Int}$ be a function of $c_\Gamma$ and $c_{interr}$, where $c_{Int} = \gamma \times c_\Gamma \times n_{Dec} + c_{interr} \times (n_C + n_{Dec})$.

---

[6]Where $n_P$ is the number of punishers (peer or pool).

[7]Where $t$ represents the trust in society, and that was defined in the previous subsection.

## 4.3. Computing payoffs

Voluntary PGGs are defined by the introduction of a non-participant or a loner strategy with a fixed payoff $\sigma$. The role of the loner is to give a chance to other strategies to invade the population. To compute the payoffs of the other strategies in voluntary PGGs, we need to take into account the probability that all other $M-1$ sampled individuals are loners. This is given by

$$P_\sigma = \frac{\binom{n_L}{M-1}}{\binom{N-1}{M-1}} \tag{4.2}$$

where $n_L$ is the number of loners in the population. $M$ is the number of agents selected to play the PGG. $N$ is the size of the population.

### 4.3.1. Cooperator payoff

$$\Pi_C = P_\sigma \times \sigma + (1 - P_\sigma) \times (\text{payout} - c) - \beta \frac{M-1}{N-1}. \tag{4.3}$$

### 4.3.2. Defector payoffs

$$\Pi_D = P_\sigma \times \sigma + (1 - P_\sigma) \times \text{payout} - \text{cost}_D \frac{M-1}{N-1}, \tag{4.4}$$

where $\text{cost}_D$ depends on the PGG that is played

  (i) Pool-punishment and peer-punishment: $n_{\text{PoP}} \times B + n_{\text{PeP}} \times b$
  (ii) Pool punishment: $n_{\text{PoP}} \times B$ or $n_{H_{\text{PoP}}} \times B$
  (iii) Peer-punishment: $n_{\text{PeP}} \times b$ or $n_{H_{\text{PeP}}} \times b$.

### 4.3.3. Pool-punishment payoffs

$$\Pi_{\text{PoP}} = P_\sigma \times \sigma + (1 - P_\sigma) \times ((\text{payout} - c) - G) + \text{reward}_{\text{PoP}} \frac{M-1}{N-1}, \tag{4.5}$$

where $\text{reward}_{\text{PoP}}$ depends on the PGG that is played

  (i) Together with peer punishers: $\beta(n_C/n_{\text{PoP}}) + n_{\text{PeP}}$
  (ii) Together with interrogators: $\beta(n_C/n_{\text{PoP}}) + n_{\text{Int}}$.

### 4.3.4. Peer-punishment payoffs

$$\Pi_{\text{PeP}} = P_\sigma \times \sigma + (1 - P_\sigma) \times (\text{payout} - c) + \text{reward}_{\text{PeP}} \frac{M-1}{N-1} - (c_b \times n_D) \frac{M-1}{N-1}, \tag{4.6}$$

where $\text{reward}_{\text{PeP}}$ depends on the PGG that is played

  (i) Together with pool punishers: $\beta(n_C/(n_{\text{PoP}} + n_{\text{PeP}}))$
  (ii) Together with interrogators: $\beta(n_C/(n_{\text{PeP}} + n_{\text{Int}}))$.

### 4.3.5. Deception payoff

$$\Pi_{\text{Dec}} = P_\sigma \times \sigma + (1 - P_\sigma) \times \text{payout} - c_{\text{Dec}} \frac{M-1}{N-1}. \tag{4.7}$$

### 4.3.6. Interrogation payoff

$$\Pi_{\text{Int}} = P_\sigma \times \sigma + (1 - P_\sigma) \times (\text{payout} - c) - c_{\text{Int}} \frac{M-1}{N-1} + \text{reward}_{\text{Int}} \frac{M-1}{N-1}, \tag{4.8}$$

where $\text{reward}_{\text{Int}}$ depends on the type of PGG:

  (i) Together with pool-punishers: $\beta \times n_C/(n_{\text{PoP}} + n_{\text{Int}})$
  (ii) Together with peer-punishers: $\beta \times n_C/(n_{\text{PeP}} + n_{\text{Int}})$.

**Definition 4.1. (cost of pool hybrid interrogators)** Let the cost of being an interrogator $c_{H_{pop}}$ be a function of $c_\Gamma$ and $c_{interr}$, where $c_{Hpop} = G + \gamma \times c_\Gamma \times n_{Dec} + c_{interr} \times (n_C + n_{Dec})$.

**Definition 4.2. (cost of peer hybrid interrogators)** Let the cost of being an interrogator $c_{H_{pep}}$ be a function of $c_b$, $c_\Gamma$ and $c_{interr}$, where $c_{Hpep} = c_b \times n_D + \gamma \times c_\Gamma \times n_{Dec} + c_{interr} \times (n_C + n_{Dec})$.

### 4.3.7. Pool-hybrid interrogation payoff

$$\Pi_{H_{PoP}} = P_\sigma \times \sigma + (1 - P_\sigma) \times ((\text{payout} - c) - G) - c_{H_{PoP}} \frac{M-1}{N-1} + \left(\beta \times \frac{n_C}{n_{H_{PoP}}}\right) \frac{M-1}{N-1}. \tag{4.9}$$

### 4.3.8. Peer-hybrid interrogation payoff

$$\Pi_{H_{PeP}} = P_\sigma \times \sigma + (1 - P_\sigma) \times (\text{payout} - c) - c_{H_{PeP}} \frac{M-1}{N-1} + \left(\beta \times \frac{n_C}{n_{H_{PeP}}}\right) \frac{M-1}{N-1}. \tag{4.10}$$

# 5. Experimental set-up

To answer our main research questions, namely RQ1 and RQ2, we model and run extensive simulations of six different PGGs with different population compositions (See electronic supplementary material, S1 appendix for details of the model).

Each simulation is a run of $10^5$ PGG games. Each game contains $N = 100$ agents. In the initial run of each simulation, the population starts with all agents being **defectors**, and after each game the population evolves as described in §4.1. Each subsequent run starts from the population composition that was generated by the previous run.

We ran two sets of simulations, one with strong social learning ($s = s \xrightarrow{\infty}$), and one with weak/ intermediate social learning ($s = 1000$). For each of the six PGGs, we ran $10^3$ simulations for each of these two learning conditions, and report results in terms of the frequencies with which agents picked particular strategies at the end of those runs. These figures are reported as averages over the $10^3$ runs.

The set-ups for all six PGGs consist of fixing the following parameter values: $M = 5$, $\mu = 0.001$, $c = 1$, $r = 3$, $\sigma = 0.3$, $b = c_b = 0.7$, $B = G = 0.7$. The fixed parameters for deception were $\beta = 0.5$, $\Gamma = 0.8$, $c_\Gamma = 0.5$, $c_{interr} = 0.5$ and commSkill = 0.5. The parameter values are identical to those used in [28] and [27], except for the $\sigma$. We used $\sigma = 0.3$ in order to incentivize participation, whereas [27,28] used $\sigma = 1$. Regarding the parameters of deception, we also tested their effects on the long-run frequencies. The six different PGGs are summarized in table 2.

Below we describe their corresponding strategy set-ups:

PGG1: This is the PGG based on [28] where second-order punishment has been substituted with a fixed tax $\beta$ that is to be paid by the cooperators for punishers to exist. This is similar to paying a tax for policing in a society. This PGG consists of cooperators, defectors, loners, peer-punishers and pool-punishers.

PGG2: In this PGG, we keep the same types of agents as in PGG1 and we introduce the deceivers. In this set-up, the deceivers are able to free-ride without risking being caught by interrogators.

PGG3: In this PGG, we keep the same set-up as in PGG2, but we replace the peer-punishers with interrogators. Interrogators are able to detect deceivers, while the pool-punishers are able to punish defectors.

PGG4: In this PGG, we keep the same set-up as in PGG2, but we replace the pool-punishers with interrogators. Interrogators are able to detect deceivers, while the peer-punishers are able to punish defectors.

PGG5: In this PGG, we keep the same set-up as in PGG3. However, instead of having two different types of agents seeking defectors and deceivers separately, we have a single type of agent that performs both jobs, namely the pool-hybrid interrogator. This is analogous to having a centralized policing institution in a society which keeps track of both types of free-riding behaviours.

PGG6: In this PGG, we keep the same set-up as in PGG4. However, instead of having two different types of agents chasing defectors and deceivers separately, we have a single type of agent that performs both jobs, namely the peer-hybrid interrogator. This is analogous to having a decentralized policing institution in a society which keeps track of both types of free-riding behaviours.

**Table 2.** PGGs and the combination of strategies.

| PGG | C | D | L | PoP | PeP | Dec | Int | $H_{PoP}$ | $H_{PeP}$ |
|-----|---|---|---|-----|-----|-----|-----|-----------|-----------|
| 1 | ✓ | ✓ | ✓ | ✓ | ✓ | ✗ | ✗ | ✗ | ✗ |
| 2 | ✓ | ✓ | ✓ | ✓ | ✓ | ✓ | ✗ | ✗ | ✗ |
| 3 | ✓ | ✓ | ✓ | ✓ | ✗ | ✓ | ✓ | ✗ | ✗ |
| 4 | ✓ | ✓ | ✓ | ✗ | ✓ | ✓ | ✓ | ✗ | ✗ |
| 5 | ✓ | ✓ | ✓ | ✗ | ✗ | ✓ | ✗ | ✓ | ✗ |
| 6 | ✓ | ✓ | ✓ | ✗ | ✗ | ✓ | ✗ | ✗ | ✓ |

The nine strategies that we used in the six PGGs are: cooperation $C$, defection $D$, non-participation (loners) $L$, peer-punishing PeP, pool-punishing PoP, deception Dec, interrogation Int, pool-hybrid interrogation $H_{PoP}$, and peer-hybrid Interrogation $H_{PeP}$
**Cooperative strategies** are the following: $C$, PoP, PeP, Int, $H_{PoP}$ and $H_{PeP}$. **Free-riding strategies** are the following: $D$ and Dec. Only one strategy, $L$, does not participate to the PGG and does not fall into any of the two previous categories.

# 6. Results

Here, we present the results from our simulations of the six PGGs with the two social learning conditions. The code for the simulations was implemented and run in Python. We report results in terms of the frequencies with which agents picked particular strategies at the end of the simulation runs for each of the six PGGs.[8] These figures are reported as averages over the $10^3$ runs.

When reading the barcharts for each PGG (figures 1–6), the coloured bars represent the long-term average frequency of agents with a given strategy where each colour represents a different strategy of the PGG. The error bars represent ±1 s.d. from this mean given that iterations of a single PGG do not necessarily have the same outcome in terms of long-run population frequencies. We also performed statistical Kruskall–Wallis non-parametric (distribution-free) tests for each PGG in order to analyse variance between payoff samples over all strategies in a given PGG assuming that strategies are dependent variables. To compare the payoff samples of the same strategies in different PGGs, we performed pairwise non-parametric Mann–Whitney tests, assuming independence between the PGGs. The results from both types of tests gave very low $p$-values ($p < 0.01$), meaning that the differences between the payoff averages obtained from our simulations are statistically significant and they have not occurred by chance.

## 6.1. PGG1: punishment

Before we could answer RQ1 and RQ2 and the related questions, we wanted to have a baseline model by reproducing the results in [28], where cooperation is established. Thus, we implemented PGG1 and we formulated the following hypotheses:

— **H1.1** *In a PGG, if the following strategies are available* C, D, L, PeP *and* PoP, *then the long-run average frequency of agents that select cooperative strategies is* **higher** *than the one of agents that select the free-riding strategies, when social learning is* **weak**. *Hence cooperation is established.*
— **H1.2** *In a PGG, if the following strategies are available* C, D, L, PeP *and* PoP, *then the long-run average frequency of agents that select cooperative strategies is* **higher** *than the one of agents that select the free-riding strategies, when social learning is* **strong**. *Hence cooperation established.*

### 6.1.1. Model assumptions

Because we wanted to reproduce the results in [28], we adopted all their assumptions except for the assumption of second-order punishment. In [28], the authors assumed that individual agents are driven by self-interest and did not consider equity or reciprocity between individuals. This assumption of *selfishness* fits with our understanding of knowledge as public good in the *Infosphere*, as argued in [33]. This assumption is also justified by the fact that individuals that interact on the web can be oblivious to others when using the knowledge resources provided by the web, without

---

[8]These frequencies are the same as the proportion of agents picking the strategies.

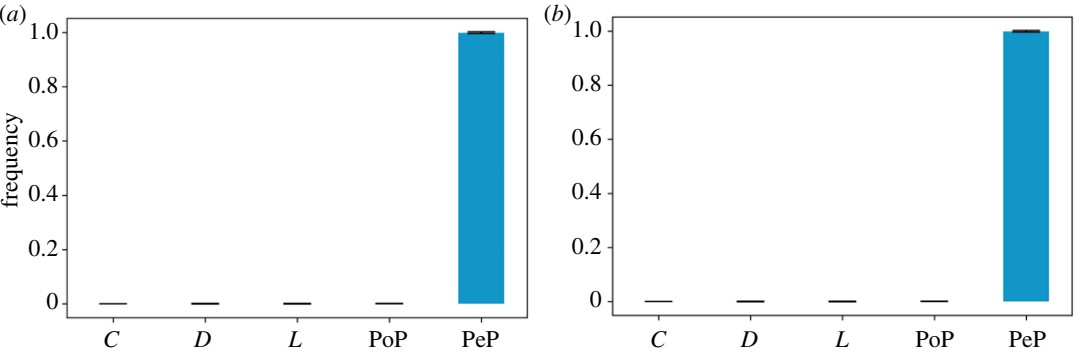

**Figure 1.** Long-run average frequencies in PGG1. The long-run average frequencies of cooperative strategies determine the establishment of cooperation due to the number of pool-punishers PeP. This is the case for weak $s = 1000$ social learning as shown in figure 1$a$ and for strong social learning $s = \xrightarrow{\infty}$ as shown in figure 1$b$. The confirmation of **H1.1** is backed by the data from the simulations illustrated in figure 1$a$. The confirmation of **H1.2** is backed by the data from the simulations illustrated in figure 1$b$. ($a$) $s = 1000$, $p < 0.01$, ($b$) $s \xrightarrow{\infty}$, $p < 0.01$.

considering the consequences of their online behaviour. Thus, PGG1 represents a system in which agents can choose between punishing defection before the PGG is played *a priori* (pool-punishment) or punishing defection after the PGG is played *a posteriori* (peer-punishment). We assume that deception is not present in this type of system.

### 6.1.2. Results

We reproduced the results in [28] without second-order punishment (figure 1$a$,$b$). Therefore, peer-punishers dominated the games with the following long-run average frequencies for: (1) $s = 1000$: [$C$ : 0.0 (s.d. = 0.0), $D$ : 0.001 (s.d. = 0.004), $L$ : 0.001 (s.d. = 0.006), PoP : 0.0 (s.d. = 0.001), PeP : 0.998 (s.d. = 0.01)]; and (2) $s \xrightarrow{\infty}$: [$C$ : 0.0 (s.d. = 0.0), $D$ : 0.0 (s.d. = 0.001), $L$ : 0.001 (s.d. = 0.003), PoP : 0.0 (s.d. = 0.0), PeP : 0.999 (s.d. = 0.004)].

## 6.2. PGG2: deception

To answer RQ1 and its related questions, we implemented PGG2 and we formulated the following hypotheses:

— **H2.1** *In a PGG, if the following strategies are available* C, D, L, PeP, PoP *and* Dec, *then the long-run average frequency of agents that select cooperative strategies is* **lower** *than the one of agents that select the free-riding strategies, when social learning is* **weak**. *Hence cooperation is destabilized.*
— **H2.2** *In a PGG, if the following strategies are available* C, D, L, PeP, PoP *and* Dec, *then the long-run average frequency of agents that select cooperative strategies is* **lower** *than the one of agents that select the free-riding strategies, when social learning is* **strong**. *Hence cooperation is destabilized.*

### 6.2.1. Model assumptions

We introduced deceivers to check if they destabilize cooperation by reducing the long-run frequency of peer-punishers. In this PGG2 model, we keep all of the assumptions from PGG1. However, we introduce the deceiver strategy to which we also apply the assumption of self-interest which is consistent with the literature, that defines deception as a selfish behaviour that aims to maximize one's payoff [11]. We also assume that in this model there is no strategy for punishing deception, that is in order to test if deceivers emerge in systems where no punishment for deception exists, and that deceptive behaviour can be simply adopted by the agent population through social learning. In this model, the cost of deception is only influenced by the *cognitive load* (See *cogLoad* in 'Modelling deception and interrogation' section). The reason behind this is that there are no agents that could interrogate and punish deceivers in this model. However, deceivers have a cognitive load because they need to be able to successfully deceive without being caught. This means, that the more players are involved in the PGG, the higher the cognitive load of the deceivers.

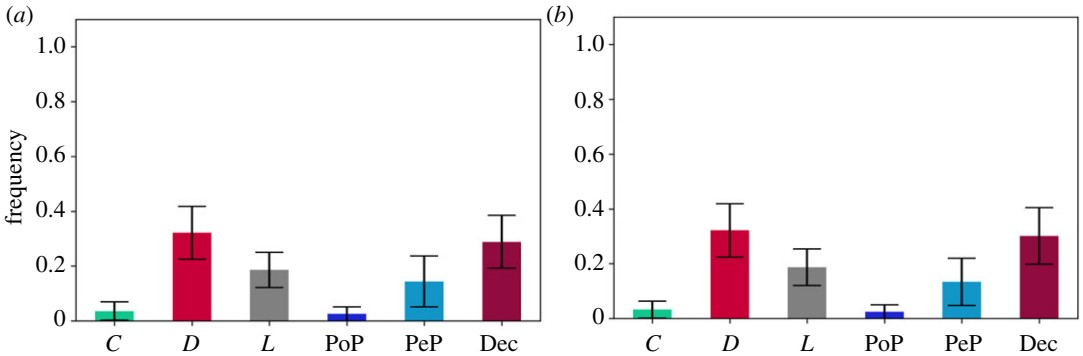

**Figure 2.** Long-run average frequencies in PGG2. The long-run average frequencies of free-riding strategies determine the destabilization of cooperation due to the higher number of *D* and Dec compared with the other cooperative strategies. This is the case for weak *s* = 1000 social learning as shown in figure 2*a* and for strong social learning $s = \overset{\infty}{\longrightarrow}$ as shown in figure 2*b*. The confirmation of **H2.1** is backed by the data from the simulations illustrated in figure 2*a*. The confirmation of **H2.2** is backed by the data from the simulations illustrated in figure 2*b*. (*a*) *s* = 1000, *p* < 0.01, (*b*) $s \overset{\infty}{\longrightarrow}$, *p* < 0.01.

### 6.2.2. Results

Deceivers have indeed had an impact on the system (figure 2*a*,*b*), given the following long-run average frequencies for: (1) *s* = 1000: [*C* : 0.032 (s.d. = 0.031), *D* : 0.324 (s.d. = 0.1), *L* : 0.187 (s.d. = 0.065), PoP : 0.026 (s.d. = 0.029), PeP : 0.133 (s.d. = 0.089), Dec : 0.295 (s.d. = 0.104)]; and (2) $s = \overset{\infty}{\longrightarrow}$: [*C* : 0.034 (s.d. = 0.031), *D* : 0.324 (s.d. = 0.099), *L* : 0.186 (s.d. = 0.064), PoP : 0.025 (s.d.=0.025), PeP : 0.136 (s.d. = 0.094), Dec : 0.294 (s.d. = 0.1)].

## 6.3. PGG3: interrogation with pool-punishment

To answer RQ2 and its related questions, we implemented PGG3 and we formulated the following hypotheses:

— **H3.1** *In a PGG, if the following strategies are available* C, D, L, PoP, Dec *and* Int, *then the long-run average frequency of agents that select cooperative strategies is* **higher** *than the one of agents that select the free-riding strategies, when social learning is* **weak**. *Hence cooperation is re-established.*
— **H3.2** *In a PGG, if the following strategies are available* C, D, L, PoP, Dec *and* Int, *then the long-run average frequency of agents that select cooperative strategies is* **higher** *than the one of agents that select the free-riding strategies, when social learning is* **strong**. *Hence cooperation is re-established.*

### 6.3.1. Model assumptions

In order to try and re-establish cooperation, we replaced the peer-punishers with interrogators in the population composition to check whether interrogation and pool-punishment have a positive impact on the system. In this PGG3 model, we keep the assumption of self-interested agents. We also keep the deceiver strategy from PGG2. However, now the cost for deception increases because in addition to *cogLoad* there is the *leakage* factor that contributes to its cost. The *leakage* is now present due to the fact that we introduce the interrogation strategy. We assume that PGG3 represents a system in which agents can choose to interrogate and punish deceptive behaviour after a PGG is played, but they can also choose to punish defection before the PGG is played. Another assumption is that the agents that interrogate and punish deceivers are not the same as the ones that punish defectors. This means that different institutions are responsible for regulating the two types of free-riding behaviour.

### 6.3.2. Results

Unfortunately, even though interrogators are present and they are able to reduce the frequency of deceivers for strong imitation, more defectors seem to be invading the system (figure 3*a*,*b*). This indicates the ineffectiveness of the pool-punishers in this PGG, as shown in the long-run average frequencies for: (1) *s* = 1000: [*C* : 0.083 (s.d. = 0.053), *D* : 0.378 (s.d. = 0.103), *L* : 0.22 (s.d. = 0.067), PoP : 0.075 (s.d. = 0.059), Int : 0.029 (s.d. = 0.027), Dec : 0.215 (s.d. = 0.107)]; and (2) $s = \overset{\infty}{\longrightarrow}$: [*C* : 0.082 (s.d. =

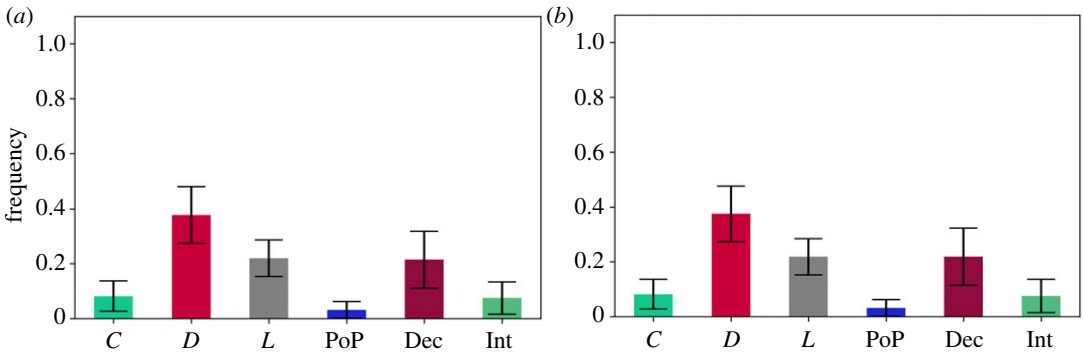

**Figure 3.** Long-run average frequencies in PGG3. The long-run average frequencies of free-riding strategies are higher than the ones of cooperative strategies. This is the case for weak $s = 1000$ social learning as shown in figure 3*a* and for strong social learning $s = \overset{\infty}{\longrightarrow}$ as shown in figure 3*b*. The falsification of **H3.1** is backed by the data from the simulations illustrated in figure 3*a*. The falsification of **H3.2** is backed by the data from the simulations illustrated in figure 3*b*. (*a*) $s = 1000$, $p < 0.01$, (*b*) $s \overset{\infty}{\longrightarrow}$, $p < 0.01$.

0.054), $D$ : 0.383 (s.d. = 0.104), $L$ : 0.219 (s.d. = 0.07), PoP : 0.076 (s.d. = 0.063), Int : 0.029 (s.d. = 0.027), Dec : 0.211 (s.d. = 0.105)].

## 6.4. PGG4: interrogation with peer-punishment

To answer RQ2 and its related questions, we implemented PGG4 and we formulated the following hypotheses:

— **H4.1** *In a PGG, if the following strategies are available* C, D, L, PeP, Dec *and* Int, *then the long-run average frequency of agents that select cooperative strategies is* **higher** *than the one of agents that select the free-riding strategies, when social learning is* **weak**. *Hence cooperation is re-established.*
— **H4.2** *In a PGG, if the following strategies are available* C, D, L, PeP, Dec *and* Int, *then the long-run average frequency of agents that select cooperative strategies is* **higher** *than the one of agents that select the free-riding strategies, when social learning is* **strong**. *Hence cooperation is re-established.*

### 6.4.1. Model assumptions

Due to the lack of efficiency of pool-punishers given by the results in both PGG1 (without deception) and PGG3 (with deception and interrogation) we decided to replace them with peer-punishers given their success in PGG1. We otherwise keep the same assumptions as in PGG3. We assume that PGG4 represents a system in which agents can choose to interrogate and punish deceptive behaviour after a PGG is played, but they can also choose to punish defection after the PGG is played. The same assumption as in PGG3 is kept, which is that the agents that interrogate and punish deceivers are not the same as the ones that punish defectors. This means that different institutions are responsible for regulating the two types of free-riding behaviour.

### 6.4.2. Results

Unfortunately, peer-punishers proved to be almost as inefficient as pool-punishers given the following long-run average frequencies for: (1) $s = 1000$: [$C$ : 0.035 (s.d. = 0.03), $D$ : 0.343 (s.d. = 0.101), $L$ : 0.196 (s.d. = 0.067), PeP : 0.118 (s.d. = 0.082), Int : 0.05 (s.d. = 0.052), Dec : 0.258 (s.d. = 0.105)]; and (2) $s = \overset{\infty}{\longrightarrow}$: [$C$ : 0.036 (s.d. = 0.033), $D$ : 0.349 (s.d. = 0.1), $L$ : 0.197 (s.d. = 0.068), PeP : 0.116 (s.d. = 0.086), Int : 0.051 (s.d. = 0.052), Dec : 0.251 (s.d. = 0.101)]. Even though peer-punishers alone perform better in the PGG, they have a negative impact on the performance of both interrogators and cooperators (figure 4*a*,*b*).

## 6.5. PGG5: pool-punishment and interrogation hybrid

To answer RQ2 and its related questions, we implemented PGG5 and we formulated the following hypotheses:

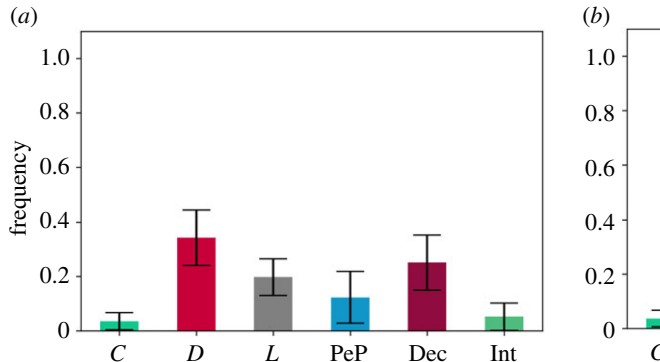
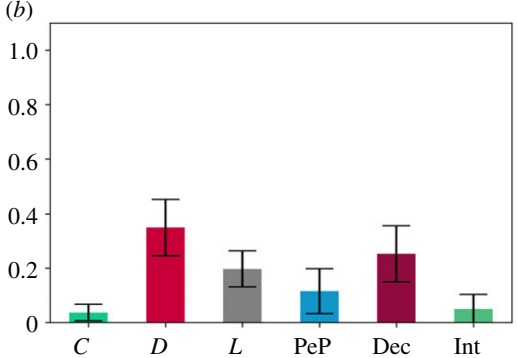

**Figure 4.** Long-run average frequencies in PGG4. The long-run average frequencies of free-riding strategies are higher than the ones of cooperative strategies. This is the case for weak $s = 1000$ social learning as shown in figure 4a and for strong social learning $s = \xrightarrow{\infty}$ as shown in figure 4b. The falsification of **H4.1** is backed by the data from the simulations illustrated in figure 4a. The falsification of **H4.2** is backed by the data from the simulations illustrated in figure 4b. (a) $s = 1000$, $p < 0.01$, (b) $s \xrightarrow{\infty}$, $p < 0.01$.

— **H5.1** *In a PGG, if the following strategies are available* C, D, L, Dec *and* $H_{PoP}$, *then the long-run average frequency of agents that select cooperative strategies is* **higher** *than the one of agents that select the free-riding strategies, when social learning is* **weak**. *Hence cooperation is re-established.*
— **H5.2** *In a PGG, if the following strategies are available* C, D, L, Dec *and* $H_{PoP}$, *then the long-run average frequency of agents that select cooperative strategies is* **higher** *than the one of agents that select the free-riding strategies, when social learning is* **strong**. *Hence cooperation is re-established.*

### 6.5.1. Model assumptions

Due to their joint inefficiency in re-establishing cooperative behaviour in PGG4, we decided to remove both interrogators and peer-punishers from the system, and we introduced pool-hybrid interrogators instead. In the PGG5 model, we adopt the assumptions from PGG3. However, instead of having two separate strategies for pool-punishment and interrogation, we introduce a single hybrid strategy to do so, namely $H_{PoP}$. The cost of deception is now influenced by the presence of $H_{PoP}$ agents. The assumption that we now make is that PGG5 is a system in which the agents that punish both deceptive and defective behaviour are part of the same coalition. This means that the same institution is responsible for regulating both types of free-riding behaviour. This institution represents the implementation of a centralized regulatory system due to its pool-punishment mechanism. However, this is a hybrid centralization because the interrogation mechanism investigates potential deceivers and punishes them individually after the PGG has been played.

### 6.5.2. Results

This type of hybrid proved to be even less efficient against deceivers and defectors than when we had both interrogators and pool-punishers acting separately (figure 5a,b). This is reflected in the long-run average frequencies for: (1) $s = 1000$: [C : 0.085 (s.d. = 0.049), D : 0.344 (s.d. = 0.091), L : 0.259 (s.d. = 0.07), $H_{PoP}$ : 0.055 (s.d. = 0.043), Dec : 0.258 (s.d. = 0.094)]; and (2) $s = \xrightarrow{\infty}$: [C : 0.082 (s.d. = 0.051), D : 0.345 (s.d. = 0.089), L : 0.263 (s.d. = 0.074), $H_{PoP}$ : 0.052 (s.d. = 0.04), Dec : 0.258 (s.d. = 0.096)].

## 6.6. PGG6: peer-punishment and interrogation hybrid

To answer RQ2 and its related questions, we implemented PGG6 and we formulated the following hypotheses:

— **H6.1** *In a PGG, if the following strategies are available* C, D, L, Dec *and* $H_{PeP}$, *then the long-run average frequency of agents that select cooperative strategies is* **higher** *than the one of agents that select the free-riding strategies, when social learning is* **weak**. *Hence cooperation is re-established.*
— **H6.2** *In a PGG, if the following strategies are available* C, D, L, Dec *and* $H_{PeP}$, *then the long-run average frequency of agents that select cooperative strategies is* **higher** *than the one of agents that select the free-riding strategies, when social learning is* **strong**. *Hence cooperation is re-established.*

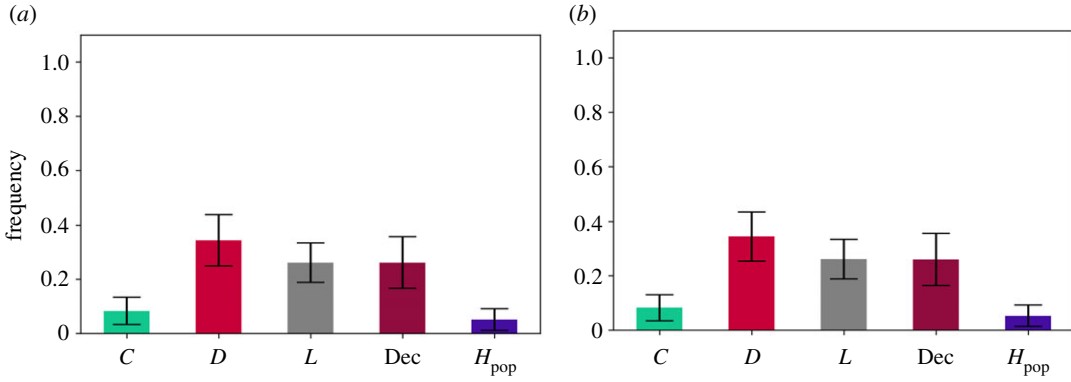

**Figure 5.** Long-run average frequencies in PGG5. The long-run average frequencies of free-riding strategies are higher than the ones of cooperative strategies. This is the case for weak $s = 1000$ social learning as shown in figure 5a and for strong social learning $s = \overset{\infty}{\longrightarrow}$ as shown in figure 5b. The falsification of **H5.1** is backed by the data from the simulations illustrated in figure 5a. The falsification of **H5.2** is backed by the data from the simulations illustrated in figure 5b. (a) $s = 1000$, $p < 0.01$, (b) $s \overset{\infty}{\longrightarrow}$, $p < 0.01$.

### 6.6.1. Model assumptions

In the PGG6 model, we adopt the assumptions from PGG4. However, instead of introducing two separate strategies for peer-punishment and interrogation, we have a single hybrid strategy to do both jobs, namely $H_{PeP}$. The cost of deception is now influenced by the presence of $H_{PeP}$ agents. The assumption that we make is that PGG6 is a system in which the agents that punish both deceptive and defective behaviour are part of the same coalition. This means that the same institution is responsible for regulating both types of free-riding behaviour. However, compared with PGG5, this institution represents the implementation of a decentralized regulatory system due to its peer-punishment mechanism. In this case, the hybridization of the mechanism with interrogation still gives a fully decentralized system.

### 6.6.2. Results

In PGG6, we replaced the pool-hybrid interrogators with peer-hybrid interrogators (figure 6a,b). For intermediate imitation $s = 1000$, peer-hybrid punishers perform much better than the pool-hybrid interrogators, but not enough to re-establish strong levels of cooperation in the system. However, for strong imitation peer-hybrid interrogators seem to re-establish strong levels of cooperation by significantly reducing the influence of defection and more importantly, deception for strong imitation. These results are reflected in the long-run average frequencies for: (1) $s = 1000$: [$C$ : 0.095 (s.d. = 0.049), $D$ : 0.295 (s.d. = 0.095), $L$ : 0.228 (s.d. = 0.076), $H_{PeP}$ : 0.184 (s.d. = 0.137), Dec : 0.199 (s.d. = 0.095)]; and (2) $s = \overset{\infty}{\longrightarrow}$: [$C$ : 0.028 (s.d. = 0.037), $D$ : 0.096 (s.d. = 0.117), $L$ : 0.07 (s.d. = 0.082), $H_{PeP}$ : 0.742 (s.d. = 0.279), Dec : 0.063 (s.d. = 0.085)].

## 6.7. Cooperation versus free-riding

The effect of the above experiments is to have compared the long-run average frequencies of the two meta-strategies, namely cooperation and free-riding, given weak/intermediate (see table 3 and figure 8a) and strong social learning (see table 4 and figure 8b). The results indicate the following:

   (i) The introduction of deception promotes free-riding in voluntary PGGs with punishment (figure 7a).
   (ii) Strong social learning does not promote cooperation when deception is present unless peer-hybrid interrogation is introduced (figure 7b).

As for the parameters that directly influence deception as a strategy in PGGs with and without peer-hybrid interrogators there are several observations to be considered that could shed light on the questions derived from RQ1 and RQ2 in §3. First, it seems that in the absence of any type of interrogation (PGG2) as well as in the presence of peer-hybrid interrogators (PGG6), deception becomes the evolutionary stable strategy if communicative skill is maximized (commSkill = $\overset{1}{\longrightarrow}$) (see electronic supplementary material, figure S6). In other words, deception is optimal if you are good at it. Second, the tax paid by cooperators for punishers and interrogators to exist influences PGG2 and PGG6 in opposite ways in terms of total cooperation. In PGG2, increases in $\beta$ promote deception and

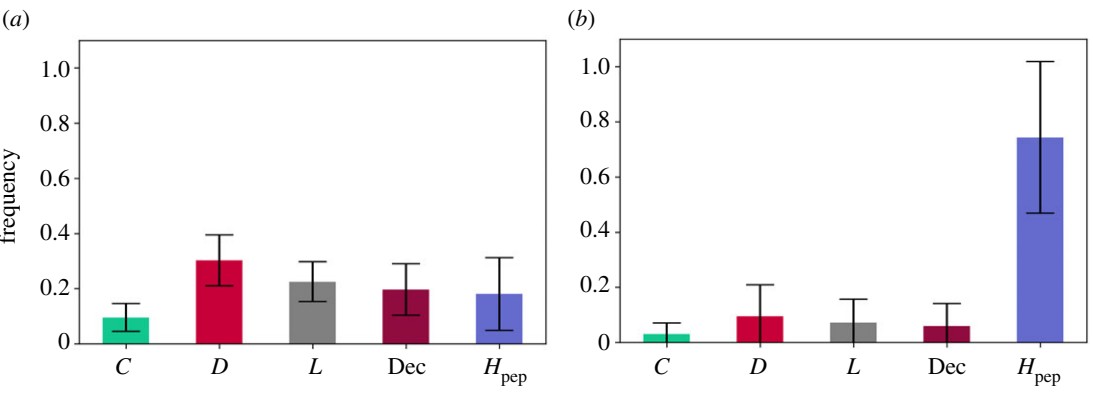

**Figure 6.** Long-run average frequencies in PGG6. The long-run average frequencies of free-riding strategies are higher than the ones of cooperative strategies for weak social learning $s = 1000$, as seen in figure 6a. This is not the case for strong $s = \xrightarrow{\infty}$ social learning where the long-run average frequencies of cooperative strategies driven by $H_{PeP}$ are higher than the ones for free-riders, as seen in figure 6b. The falsification of **H6.1** is backed by the data from the simulations illustrated in figure 6a. The confirmation of **H6.2** is backed by the data from the simulations illustrated in figure 6b. (a) $s = 1000$, $p < 0.01$, (b) $s \xrightarrow{\infty}$, $p < 0.01$.

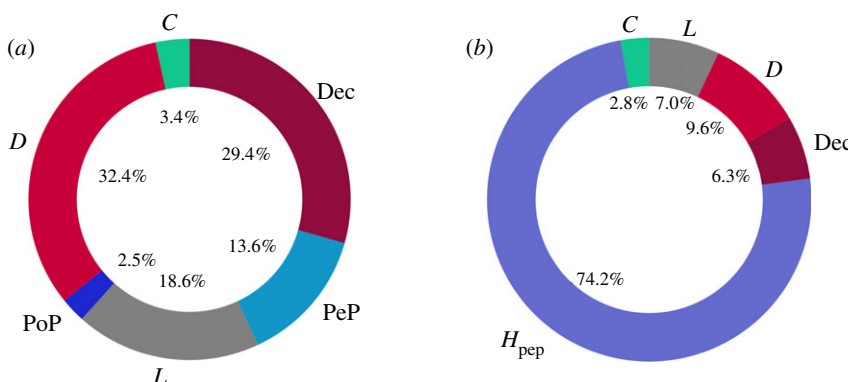

**Figure 7.** Long-run average frequencies in PGG2 versus PGG6 for $s \xrightarrow{\infty}$, for which $D$ and Dec frequencies are significantly different between the two PGGs $p < 0.01$. (a) PGG2 and (b) PGG6.

**Table 3.** Cooperation versus free-riding long-run frequencies for weak/intermediate social learning.

| PGG | 1[a] | 2 | 3 | 4 | 5 | 6 |
|---|---|---|---|---|---|---|
| cooperators | 0.99 | 0.193 | 0.187 | 0.202 | 0.139 | 0.278 |
| free-riders | 0.0 | 0.618 | 0.592 | 0.601 | 0.601 | 0.493 |

[a]For weak/intermediate levels of social learning, free-riding dominates in all PGGs where deception is present.

by extension free-riding, while in PGG6, increases in $\beta$ promote peer-hybrid interrogation. Thirdly, increases in the tax on deception $\Gamma$, which is inflicted by all the interrogators in all PGGs where these are present, have a positive impact in all PGGs. The increase manages to significantly reduce the frequency of deceivers in all PGGs where it can be inflicted; however, it only has a positive impact on total cooperation in PGG5, where it promotes pool-hybrid interrogation and cooperation, and in PGG6 where it promotes peer-hybrid interrogation. The drawback is that in PGG5, $\Gamma$ needs to be very high ($\Gamma > 800$) in order to even begin promoting the cooperative strategies (see electronic supplementary material, figure S6). This is not the case for PGG6, where increases in $\Gamma$ have a considerable impact on promoting cooperation as a meta-strategy (see electronic supplementary material, figure S4).

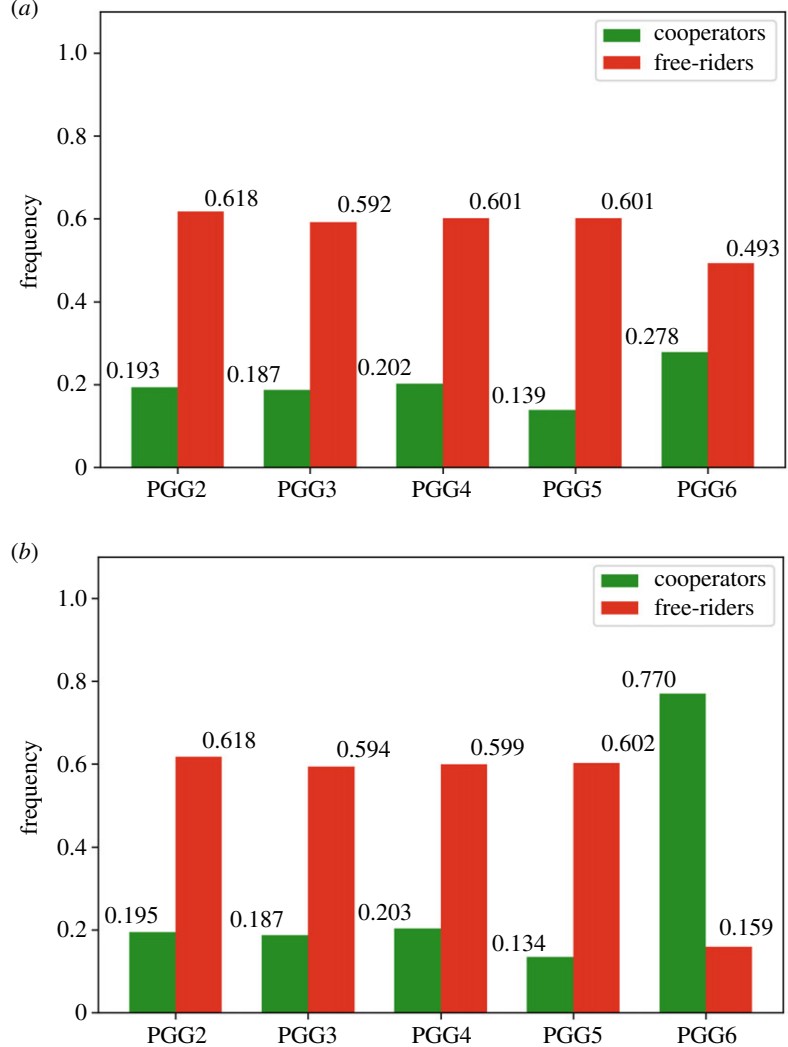

**Figure 8.** Long-run average frequencies for meta-strategies: cooperation versus free-riding. (*a*) $s = 1000$, (*b*) $s \xrightarrow{\infty}$.

**Table 4.** Cooperation versus free-riding long-run frequencies for strong social learning.

| PGG | 1[a] | 2 | 3 | 4 | 5 | 6 |
|---|---|---|---|---|---|---|
| cooperators | 0.99 | 0.195 | 0.187 | 0.203 | 0.134 | 0.77 |
| free-riders | 0.0 | 0.618 | 0.594 | 0.599 | 0.602 | 0.159 |

[a]For strong levels of social learning, free-riding dominates in PGGs where deception is present, but cooperation is re-established when peer-hybrid Interrogation is introduced.

## 7. Discussion

Can societies reach cooperation in systems of public knowledge where deception is present? Our results show two possible outcomes. One possible outcome is that cooperation between agents cannot reach high levels in the case of weak or intermediate social learning (imitation strength) (PGGs 2,3,4,5). However, cooperation is slightly promoted if there is a hybrid and decentralized regulatory institution (or system, e.g. the Internet) that allows the punishment of defectors using decentralized methods (which is represented by peer-punishing) and that allows the independent interrogation and peer-punishment of deceivers (PGG6). The other possible outcome happens in the case of strong social learning, where cooperation between agents can be reached when such a decentralized regulatory system is present (PGG6) (see electronic supplementary material, figure S2). What this means in the

real world is that agents need to quickly learn how to identify and punish social sources of deception. The effect of strong imitation shows us that if agents learn to quickly adopt other agent's strategies to form a coalition, then free-riding can be suppressed and cooperation re-established if there exists a system that allows the peer-punishment of defectors and a decentralized interrogation of potential deceivers. Strong social learning, however, does have its downsides if deception is not investigated and punished, such as in PGG2 where deceivers can become dominant (see electronic supplementary material, figure S1).

If the regulatory institution is hybrid, but it is centralized and employs pool-punishment for defection, cooperation can become a viable strategy only at very high costs such as a very high tax for the regulatory institution to exist (PGG5). On a similar note, increases in taxes for maintaining a regulatory institution which does not investigate and punish deceptive behaviour is highly detrimental to cooperation as it promotes deceptive behaviour (PGG2). It is also very likely that cooperation cannot be established if malicious agents are highly skilled at communicating deceptively independently of the type of regulatory institution.

Similar conclusions have been drawn in [27] in the case of *corruption*. Corruption is able to break down cooperation in societies where centralized institutions perform regulation. Decentralized regulatory institutions have also proven to be much more efficient in re-establishing cooperation when corruption is present.

Recently, the European Union (EU) has taken a controversial stance on data sharing [36]. The EU believes that in order for it to be able to overtake the US in terms of technological progress, then it must incentivize market fairness and technological competition between big and small businesses that own or have access to user data. In order to create this incentive, it must implement and regulate a system in which knowledge is shared between businesses, e.g. in which knowledge is becoming a public good. This is an equivalent of a single market for data. There are major concerns whether businesses will voluntarily commit to contribute to this public data pool or not, whether they stick to their commitment or not, but also whether businesses that commit to contribute would actually contribute with data that they know is truthful or genuine. Can businesses as social knowledge sharing agents reach cooperation or will they decide to free-ride on the public good? What kind of regulation mechanism should the EU implement for ensuring that businesses will cooperate? If future businesses cooperate, then how do they ensure the cooperation of their users, human or artificial, in terms of data sharing? A similar problem has emerged in the COVID-19 crisis with the issue of deploying governmental apps for tracking the symptoms and movement of citizens, in order to manage the pandemic.

From the perspective of a PGG as a knowledge sharing activity between social agents, we can conclude from the results of our PGG models that deceptive information can break down cooperation, and, by extension, the trust between agents through the promotion of defection. This results in agents adopting a free-riding strategy either by not sharing information at all (defection), or by sharing deceptive information, that is similar to sharing *fake news* or *fake data* if we want to use contemporary terms.[9] However, the impact of *fake news* or *fake data* can be mitigated if regulatory measures are taken. In the case of knowledge sharing, peer-hybrid interrogation would represent regulating knowledge sharing through the following: (i) the case-by-case demotion of individuals that make use of the knowledge pool, but do not contribute to it (defection); (ii) the case-by-case interrogation, and where necessary, demotion of knowledge sources (social agents in this case). This suggests that deception can be countered, but the challenge is to identify what mechanisms in the real world can play the role of peer-hybrid interrogation.[10]

Not addressing the issue of deception through regulation in the sharing of public knowledge can lead to bleak outcomes in hybrid societies. In [33], the authors discuss what they call the *Tragedy of the Digital Commons* (TDC), that represents Hardin's *Tragedy of the Commons* applied to the digital sphere of information and knowledge sharing. An important concept presented by the authors in [33] is the one of *exploitation* and *information pollution* of the *Infosphere*.[11] Knowledge exploitation and pollution can be caused by the deceptive behaviour of agents or coalitions of agents, human or artificial. In terms of PGG models, we can assume that the deceivers exploit the public knowledge (what is shared on the

---

[9]Speaking of fake news, another form of deception is to call out truthful news as fake news, a deceptive strategy that has become increasingly popular among some contemporary political figures.

[10]One might hope that this role would be played by the media, but, to continue our commentary on the political zeitgeist in which this paper was written, the media has largely failed in its duty in this regard.

[11]For example, the cyberspace. However, the Infosphere is not limited to online environments, see [37] for a detailed description of the Infosphere.

Infosphere publicly) by accessing and using the information that is publicly available, while also pretending to contribute to this public knowledge. Remember that our deceivers pretend to be cooperators. However, the information deceivers contribute with can be considered untruthful (fake news, forged knowledge etc.). The advancement of AI could lead to increase the risk of TDC, as machines that have the necessary capabilities to deceive and learn from social interactions, will eventually adopt deceptive knowledge sharing behaviour to better adapt as agents of a society. Social media platforms which centrally regulate the publicly shared knowledge, such as Facebook and Instagram, are systems in which deceptive behaviour (even of simplistic artificial agents) easily emerges, as we have seen for some time [38].

Fortunately, the results of this paper's PGG simulations indicate that, from an evolutionary perspective, TDC can be avoided in the case of deception if the Infosphere is regulated in a decentralized manner that organizes the public knowledge in such a way as to allow agents to voluntarily investigate each other and the information that they share publicly (see PGG6). A real-world example of such a system was the user interaction protocol implemented by Silk Road on the Dark Web [39]. The Silk Road implemented a reputation mechanism through their discussion forums for users to publicly check what information (for instance descriptions of products sold by users) they have previously shared, as well as if the information was indeed genuine, and how their past interactions have turned out. Members were even rewarded for finding out bad vendors on the platform, which can represent the behaviour of our peer-hybrid interrogators. The quality of the reviews as well as the quality of information standards of the Silk Road community, unfortunately propelled it to the undisputed best platform for drug dealing. Research on reputation mechanisms in social multi-agent systems has shown that reputation mechanisms allow agents to form a model of trust of other agents by looking at their past behaviour (what they have previously communicated), which, of course, needs to be observable (public) [40].

A similar initiative to the Silk Road with respect to the knowledge as a public good has been started by the founder of Wikipedia, Jimmy Wales, albeit for higher moral purposes such as helping society fight fake news and not be solely reliant on reputation mechanisms of its members. The initiative was initially launched through WikiTribune [41]. WikiTribune was a news wiki that used crowd-funding to financially support the costs of running a small team of professional journalists that were intended to work collaboratively with voluntary expert citizens to find stories, create content and fact-check its own work. However, according to Wales, due to issues in the design of the website, WikiTribune had failed to make its community flourish. That is when Wales turned the initiative into a microblogging and social media platform named WT.Social (WikiTribune Social) [42], arguing that the WT.Social could succeed where WikiTribune had failed. WT.Social aims to promote high-quality content and debate among its users, and its format is meant to combat fake news by providing evidence-based news with links and clear sources. The service is advertisement and click-bait free, and runs off donations from its users, similarly to Wikipedia. Unlike the other social media platforms, where users need to first report offensive content and only after the company would eventually decide to remove the reported content, in the case of WT.Social, the community is encouraged to take down material perceived to be violating the network's standards.

The philosophical and political concept under which platforms such as WT.Social aim to promote is called deliberative democracy [43], from which what Habermas calls the *public sphere* emerges [44]. The public sphere represents a fertile ground from which public opinions are formed through knowledge sharing. A public sphere that works in an ideal manner represents the foundation on which mediation takes place between state (regulator) and society which permits democratic control of state activities. For the public sphere to work in an ideal manner, a society must keep a record of state-related activities and legal actions which is publicly accessible in order to allow discussions and the formation of a public opinion. In our current era, the public sphere has become increasingly digitized. Due to this digitization, e.g. through the digitization of news and emergence of social media platforms, the formation of public opinion has been both accelerated and scaled up due to the increasing communicative means and styles that could be employed to reach an increasing number of public members. It has become a *digital public sphere* (DPS) [45].

However, even if the DPS offers more possibilities of communicating and sharing knowledge, it has certainly failed to adhere to principles of rationality and civility proposed by deliberative democracies [46], mainly due to information pollution produced by fake news. The emergence of fake news has enhanced the visibility of the DPS's weaknesses, but unfortunately it has also enhanced its negative effects on the formation of public opinion, making it susceptible to the TDC despite the recent efforts made by projects such as WT.Social and WikiTribune to mitigate these effects. Moreover, the

escalating hybridization between human and artificial societies increases the risk of propagating these effects even further, through the development of autonomous artificial agents that not only possess the ability to meaningfully engage in deliberation, but that also possess deceptive intent. Such advancements in AI would imply going beyond the current threat of AI bots and tools used by human agents for fake news propagation and generation, which are based on machine learning techniques. These are merely tools in the hands of human agents, and these tools do not possess deceptive intent. What we are referring to are neither AI tools nor artificial agents that just learn a deceptive policy and mindlessly apply it, but we refer to artificial agents that are able to truly engage in deliberation on the digital public sphere. These artificial agents could perform complex reasoning and apply it to decision-making such that they form their own goals and intentions which they act upon, and by doing this, they could eventually out-think and out-smart humans and other artificial agents when interacting in the public sphere. Remember that our model suggests that even for systems such as PGG6, where cooperation can be re-established in public knowledge sharing systems where deception is present, if the communicative skill of deceptive agents is high (commSkill $= \xrightarrow{1}$), then the system fails to promote cooperation. Deception in the DPS would then evolve, such that it would become *de-anthropomorphized*, as human agents would not be the only agents with deceptive intent and capable of truly deceiving others.

Perhaps a future solution to aid the moderation and content checking of platforms such as WT.Social will emerge from where their potential difficulties will arise, namely the further advancement of AI. However, it would not be sufficient to just advance AI deception detection by tweaking truth-bias and scepticism levels to detect deception as it is currently done in verbal and non-verbal cue-based deception detection AI research [47]. Cue-based approaches in AI deception research could potentially lead to what is called in the psychology of deception *confirmation bias* [48]. To illustrate the confirmation bias, Bond and Fahey describe the notorious *Othello error* in [48]. Othello believed that Desdemona, his wife, was cheating on him with another man. When Desdemona negated the fact that she was cheating on him, Othello did not believe her and he also exhibited suspicion that he was not believing her. Because Desdemona perceived that Othello did not believe her, Desdemona became desperate and started crying, thus she exhibited behaviour that correlates with a cheating wife. Desdemona became desperate because she believed that no matter what counter-arguments and evidence she would offer Othello, he would only take into consideration the information that would confirm his hypothesis that she was cheating on him. Desdemona was then killed by Othello because she behaved like a person who is desperate. Othello's fallacy was that he took into consideration only the behaviour a guilty person would exhibit, without taking into consideration all the other cues that might have falsified his beliefs, such as the fact that desperation causes individuals to exhibit some of the behaviours a guilty person would exhibit. In the case of complex reasoning artificial agents, Sarkadi *et al.* have shown in [24] that high levels of scepticism in communicative social interactions between artificial agents could lead to deception even when the deceiver agent's communicative skill is low. This type of artificial agent deception, argue the authors in [24], represents the special case of unintended deception where the deceiver does not act deceptively because it wrongly estimates that deception would fail, but the interrogator (the deceiver's target) is so sceptical that it caused it to believe that the deceiver has attempted deception, and thus the interrogator is caused to infer something that is false from a truthful message.

A possible solution from the advancement in AI would instead be the actual development of artificial agents capable of complex reasoning to address the issue of deception. Apart from the potential risk of being capable of deception themselves, complex reasoning artificial agents could play the roles of editors and investigative journalists (or to assist or engage with humans that fulfil these roles) which edit and moderate social networking platforms, as well as interact with users to produce high-quality content to increase the public's knowledge and to mediate the formation of public opinion. These artificial agents could potentially neutralize deceptive ones by matching their communicative capabilities, e.g. keeping commSkill < 1 in a PGG6-type of system. However, in order for artificial agents to be able to perform these roles that are beneficial to society and cooperation, much needs to be done in terms of AI research. To do so, we first need to enable artificial agents to understand deception (it takes one to know one) by successfully engaging in social interactions. We must mainly enable them: to form and to reason about arguments, to explain reasons behind decisions such that they can be held accountable by the community, and to engage in meaningful dialogue with the community and its members in a democratic manner. The areas of argumentation, human–agent interaction, explainable AI, and multi-agent systems will prove to be crucial in the future research and development of these types of artificial agents.

Finally, we note that another interesting line of future work would be to explore the connection between our model and the empirical frequency of human-to-human deception [49], and perhaps

calibrating the model in order to match the respective data. This could potentially help us to use our agent-based approach to understand what the values of the model's parameters are in face-to-face human-to-human interactions where data shows that only a few deceptive agents are responsible for most of the lies.

# 8. Conclusion

In this paper, we have presented an approach inspired by the machine behaviour movement [25]. This study shows a proof of principle of (i) how deception can emerge and destabilize cooperation in societies where centralized and decentralized regulatory institutions/systems exist; and (ii) how cooperation can be re-established in such societies. Moreover, the results inform us that there are indeed risks of machines to adopt deceptive behaviour from social interactions with other agents, enhancing the negative effects that lead to a Tragedy of the Digital Commons (TDC). However, this research also points towards a potential solution to avoid a TDC that comes from (i) avoiding the adoption of centralized systems for regulating public knowledge, and instead (ii) aiming for the adoption of a decentralized system for regulating knowledge as a public good in which agents can investigate and check the publicly shared knowledge as well as peer-punish the deceivers and defectors. Some real-world examples of these decentralized systems are platforms that implement reputation mechanisms, where agents can check what others have previously communicated, and platforms such as social networks where high-quality content and fact-checking are promoted, and where the source of the content is transparent (made public). These types of systems correspond to a digital public sphere in which human and artificial agents engage in knowledge sharing activities similarly to a deliberative democracy.

Data accessibility. Data are available at https://srkxx.github.io/Data-EvoDec/. The data for the reproduction of the statistical analysis are available at the Dryad Digital Repository: doi:10.5061/dryad.vdncjsxr8 [50].

Authors' contributions. Ş.S. model development, code, statistical analysis and evaluation. A.R. model development and code. P.M., S.P. and I.R. supervised the findings of this work. All authors discussed the results and contributed to the final manuscript.

Competing interests. We declare we have no competing interests.

Funding. Ş.S. has received PhD funding from King's College London and the MIT Media Laboratory. A.R. and I.R. acknowledge support from the Ethics & Governance of AI Fund. P.M. and S.P. have not received any external funding for this work.

Acknowledgements. The authors wish to thank the associate editors and the anonymous reviewers for the quality of their feedback and very insightful comments that have contributed to the final version of the paper.

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
