## [Peer Review File · Royal Society Open Science]

Review History

Decision letter (RSOS-191506.R0)

12-Dec-2019

Dear Mr Sarkadi:

Manuscript ID RSOS-191506 entitled "The Evolution of Deception" which you submitted to Royal Society Open Science, has been reviewed. The comments from reviewers are included at the bottom of this letter.

In view of the criticisms of the reviewers, the manuscript has been rejected in its current form. However, a new manuscript may be submitted which takes into consideration these comments.

Please note that resubmitting your manuscript does not guarantee eventual acceptance, and that your resubmission will be subject to peer review before a decision is made.

Once you have revised your manuscript, go to <https://mc.manuscriptcentral.com/rsos> and login to your Author Center. Click on "Manuscripts with Decisions," and then click on "Create a

Resubmission" located next to the manuscript number. Then, follow the steps for resubmitting your manuscript.

Your resubmitted manuscript should be submitted by 10-Jun-2020. If you are unable to submit by this date please contact the Editorial Office.

Kind regards,
Anita Kristiansen
Editorial Coordinator
Royal Society Open Science
openscience@royalsociety.org

on behalf of Professor Bart de Moor (Associate Editor) and Marta Kwiatkowska (Subject Editor)
openscience@royalsociety.org

Associate Editor Comments to Author (Professor Bart de Moor):

The manuscript treats an important challenge that is highly relevant for today's society that is heavily impacted by social media, in which 'fake news' has an increasingly 'disturbing' role.

While seemingly scientifically sound, the authors should do a better job in explaining how realistic and relevant all of the assumptions and simplifications are in the 6 PGG models presented, and how the results could possibly be implemented in realistic environments. While the introduction adequately refers to the current relevance of the research proposed, the conclusions in the section 'Results' are highly quantitative, but the qualitative discussion in the 'Discussion' part is rather terse, which complicates assessing the societal relevance and impact of the results. Also, it would be interesting to hear the authors opinion on how to implement the results in current mixed democratic-undemocratic socio-political systems on a global scale.

If - and this is stressed - the authors are able to address these concerns in a substantially revised manuscript, we may be able to proceed further. Nevertheless, please accept our apologies for the unusual length of time taken to reach this recommendation - the handling Associate Editor has been involved in a lengthy period of travel.

Author's Response to Decision Letter for (RSOS-191506.R0)

See Appendix A.

RSOS-201032.R0

Review form: Reviewer 1

Is the manuscript scientifically sound in its present form?

Yes

Are the interpretations and conclusions justified by the results?

Yes

Is the language acceptable?

Yes

Do you have any ethical concerns with this paper?

No

Have you any concerns about statistical analyses in this paper?

No

Recommendation?

Accept with minor revision (please list in comments)

Comments to the Author(s)

This article aims at developing our understanding of how deception emerges in a society under competitive, i.e. evolutionary, pressures. It presents a model, based on the public goods game, in which a number of variations of deceptive behaviour are explored. The primary aim of the work is to attempt to identify ways of reducing deception.

The topic of the article is an important and timely one, and its relevance is well-contextualised in terms of the role of AI and social media in phenomena like fake news. The insights from the article have the potential to help us understand how to tackle issues that arise from this.

The third paragraph of the introduction takes this a step further, and puts forward a perspective on how we ought to do this. This section, while I agreed with much of it personally, takes the step from a statement of facts towards a normative statement, for example stating that punishment is required. There are a few snuck premises in here, and a bit of a cleaner separation between the point of view offered about "what ought to be done" vs the context statements above, would be helpful.

In general, there are two main areas I will address in my review, where I think the paper would benefit from further consideration.

1) Positioning and Contextualisation of the Work

Considering the overall problem studied is one of knowledge management within the context of the evolution of cooperation, Section 2 might benefit from some further positioning of the authors' work in the context of that by other existing models of this. E.g., by Ober and collaborators:

- Burth Kurka D, Pitt J, Ober J, 2019, Knowledge management for self-organised resource allocation, ACM Transactions on Autonomous and Adaptive Systems, Vol:14

- Ober, J, 2008, Democracy and Knowledge. Princeton University Press.

Similarly, the work is clearly positioned in the domain of the establishment and sustainability of institutions for cooperation in multi-agent games (for example avoiding the tragedy of the commons), but there is little mention of the theory of institutions (e.g. Douglass North, Elinor Ostrom). Ostrom's work, in particular, has been operationalised in multi-agent systems, so would seem relevant. E.g.

- Pitt J, Schaumeier J, Artikis A, 2013, Axiomatisation of Socio-Economic Principles for Self-Organising Institutions: Concepts, Experiments and Challenges, ACM Transactions on Autonomous and Adaptive Systems

It would be useful for interested readers to be able to understand how these theoretical frameworks relate to your work.

While of course the work itself will necessarily consider deception in a way that is specific and restricted in scope (otherwise making progress would be very hard), the paper would benefit from some broader consideration of deception in some places, to avoid being misleading. For example, the statement "In game-theoretic terms, deception should be considered (as it is, by the literature) to be a non-cooperative behaviour" would not seem to me to be always true. For example, what about storytelling, surprise parties, and gift giving? These are not non-cooperative acts. Further, again the use of the word "should" here implies a normative position; is it not sufficient to say that it "is" considered this way, or that your work is concerned with the cases where deception is non-cooperative? Is there a deeper point you're trying to make, here?

2) Presentation of the Model and Experimental Design

Regarding the method chosen, this would seem to me to be an individual-based (i.e. agent-based) simulation to solve an evolutionary game theoretic model. However, despite the authors stating that it is an evolutionary game theoretic model, the setting is not described in the usual EGT sense (i.e. replicator dynamics describing changes to the proportions of traits in a population), but instead more a description of an ABM using discrete agents. Either would be fine for the purposes of this study, in my view, but some further clarity and precision in Section 3 would be welcome, in order to make the work more accessible to those coming from an EGT background.

With the methodology more explicit, a related question to this would be concerned with how the choice of methodology affects the sort of predictions that can be made. For example, in relation to the issues raised by Powers et al:

- Powers, S. T., Ekárt, A. & Lewis, P., 2018, Modelling enduring institutions: The complementarity of evolutionary and agent-based approaches, *Cognitive Systems Research*. 52, p. 67-81

Without some of the above clarifications, the work is still complete in terms of required details, but perhaps lacks some of the accessibility and insight it might otherwise provide.

Perhaps my main issue with the paper is that it presents an experimental study but without any explicit research questions or hypotheses. This makes it difficult to understand why the authors did the things they report on.

For example, when we get to the phrase "We model six different PGGs with different population compositions" I found myself just asking "why?", since this was not presented as part of an experimental design intended to answer any particular question. I think it would help to draw out hypotheses first, and then describe the experiments and how they are designed to test those hypotheses.

Similarly, when we get to the Deception Model, there are a number of psychological concepts (communicative skills, cognitive load, leakage of evidence) that are introduced and, while they may be interesting, they are not motivated by any particular hypothesis or stated research question in the paper. In general, I found myself asking "why?" several times through Section 4, as the authors described what they did.

Essentially, grounding the experimental work in a section earlier on, where research questions / hypotheses are laid out, would help to motivate and justify these design decisions. These are in fact present in the paper later on, in an implicit way, e.g.

- "We introduced Deceivers to check if they destabilise Cooperation"
- "...in order to test if Deceivers emerge in systems where no punishment for deception exists"

but the reader has to wait quite a while to get this context. In general, the way the results section is written, particularly the assumptions parts, there are a number of implicit hypotheses that could be drawn out explicitly.

In terms of the results themselves, there are a number of interesting insights here, and these are drawn out well in the Discussion section. The exposition of the possible implications of these results in the context of issues of the day is also very welcome.

Minor issues:

- There appears to be some text missing at the end of the "Interrogator" bullet on page 5.
- The paper refers (on p5) to this being Cooperative Game Theory. Is that really the case? Note that there is a difference between studying the emergence of cooperation in a competitive game, and what is usually described as Cooperative Game Theory, the latter being typically concerned with the formation of coalitions. Some clarification here would help.
- Please ensure all figures have appropriate labels (e.g. chart axes). It would also be nice to read fuller captions explaining what insight each figure provides.
- One of the claims made is that "deception becomes the optimal strategy if communicative skill is maximised". What is meant by "optimal strategy" here? Is it a dominant strategy? A Nash Equilibrium strategy? An Evolutionary Stable Strategy? Again, use of established terms in (evolutionary) game theory would help the accessibility of the paper and its results.
- On page 6, the notion of trust is suddenly introduced. Trust is quite a loaded term, and has several different interpretations in various fields. I am not convinced by what is there at the moment that trust becomes relevant here, but if it is, then there needs to be a clearer exposition of what sort of trust you are talking about here (ideally taking a definition that applies to this work). As things stand, I found the passage that begins with "Trust has the following properties in our model..." rather confusing; it appeared with very little context.

Review form: Reviewer 2

Is the manuscript scientifically sound in its present form?

Yes

Are the interpretations and conclusions justified by the results?

Yes

Is the language acceptable?

Yes

Do you have any ethical concerns with this paper?

Yes

Have you any concerns about statistical analyses in this paper?

Yes

Recommendation?

Major revision is needed (please make suggestions in comments)

Comments to the Author(s)

The reviewer's comments' on the manuscript entitled "The Evolution of Deception"

The topic of deception is very interesting in the fields of philosophy, psychology, economics and political science. This paper builds a new model combining Interpersonal Deception Theory and Truth-Default Theory, to analysis of the growth of deception in societies and the effectiveness of several approaches to reducing deception. The results of this paper indicate that cooperation in knowledge sharing can be re-established in systems by introducing institutions that investigate and regulate both defection and deception using a decentralized case-by-case strategy. The work done is interesting and the discussion seems fine. However, some issues should be addressed before the paper can be published.

Some comments are listed as follows:

- 1) The model part lacks the parameter range description and rationality analysis, that is, what are the value ranges of the parameters in each PGG? What are the dominant strategies for different value ranges? Are these settings reasonable?
- 2) The payoff functions of different strategies in every PGG are important channels for readers to understand the model proposed in this article. This should be shown in the manuscript rather than in the attachment.
- 3) The results of this paper are obtained through simulation, but there is no setting of initial conditions.
- 4) There is almost no result analysis. For example, in the results section of PGG2, the first sentence "Deceivers have indeed had an impact on the system" is followed by enumerating the data. However, how does deception affect the evolution of cooperation and how is it reflected in the data? There is no explanation. Other results have similar problems, which will make it difficult for readers to understand.
- 5) What is the role of Loner? In the article DOI: 10.1126/science.1070582, the introduction of the loner can restart cooperation and avoid the dilemma of defection. So, since loner is introduced in this article, what is its effect on the results?
- 6) The "PeP" and "PoP" in Table 1 do not match their introductions, their order may be reversed, please check carefully.
- 7) References should be cited in order.
- 8) There are nine strategies in this paper, but you said "The eight strategies that we used..." on Page7 line29.
- 9) On Page 9 line22, the titles' formats of Table 2 and Table 1 are not uniform, one has a '.' at the end, while the other does not.
- 10) Table 2 is missing header row.
- 11) The location of the table and its introduction are too far away. The table should be placed in a suitable location for readers to view.
- 12) Similarly, the result pictures and their introduction are too far apart. In order to facilitate readers' reading, the pictures should be placed near the corresponding results. Besides, the captions of the pictures are posted too close, which affects the display of the picture.
- 13) What does "sboth" at page12 line5 mean?

Decision letter (RSOS-201032.R0)

Dear Mr Sarkadi

The Editors assigned to your paper RSOS-201032 "The Evolution of Deception" have now received comments from reviewers and would like you to revise the paper in accordance with the reviewer comments and any comments from the Editors. Please note this decision does not guarantee eventual acceptance.

Please submit your revised manuscript and required files (see below) no later than 21 days from today's (ie 29-Jan-2021) date. Note: the ScholarOne system will 'lock' if submission of the revision is attempted 21 or more days after the deadline. If you do not think you will be able to meet this deadline please contact the editorial office immediately.

on behalf of Professor Bart de Moor (Associate Editor) and Marta Kwiatkowska (Subject Editor)
openscience@royalsociety.org

Associate Editor Comments to Author (Professor Bart de Moor):

Comments to the Author:

Please revise your paper to address the concerns of the reviewers - we may not be able to offer a further opportunity to revise your paper so please take care to fully engage with the comments and queries raised. Good luck.

Reviewer comments to Author:

Reviewer: 1

Comments to the Author(s)

This article aims at developing our understanding of how deception emerges in a society under competitive, i.e. evolutionary, pressures. It presents a model, based on the public goods game, in which a number of variations of deceptive behaviour are explored. The primary aim of the work is to attempt to identify ways of reducing deception.

The topic of the article is an important and timely one, and its relevance is well-contextualised in terms of the role of AI and social media in phenomena like fake news. The insights from the article have the potential to help us understand how to tackle issues that arise from this.

The third paragraph of the introduction takes this a step further, and puts forward a perspective on how we ought to do this. This section, while I agreed with much of it personally, takes the step from a statement of facts towards a normative statement, for example stating that punishment is required. There are a few snuck premises in here, and a bit of a cleaner separation between the point of view offered about "what ought to be done" vs the context statements above, would be helpful.

In general, there are two main areas I will address in my review, where I think the paper would benefit from further consideration.

1) Positioning and Contextualisation of the Work

Considering the overall problem studied is one of knowledge management within the context of the evolution of cooperation, Section 2 might benefit from some further positioning of the authors' work in the context of that by other existing models of this. E.g., by Ober and collaborators:

- Burth Kurka D, Pitt J, Ober J, 2019, Knowledge management for self-organised resource allocation, ACM Transactions on Autonomous and Adaptive Systems, Vol:14

- Ober, J, 2008, Democracy and Knowledge. Princeton University Press.

Similarly, the work is clearly positioned in the domain of the establishment and sustainability of institutions for cooperation in multi-agent games (for example avoiding the tragedy of the commons), but there is little mention of the theory of institutions (e.g. Douglass North, Elinor Ostrom). Ostrom's work, in particular, has been operationalised in multi-agent systems, so would seem relevant. E.g.

- Pitt J, Schaumeier J, Artakis A, 2013, Axiomatisation of Socio-Economic Principles for Self-Organising Institutions: Concepts, Experiments and Challenges, ACM Transactions on Autonomous and Adaptive Systems

It would be useful for interested readers to be able to understand how these theoretical frameworks relate to your work.

While of course the work itself will necessarily consider deception in a way that is specific and restricted in scope (otherwise making progress would be very hard), the paper would benefit from some broader consideration of deception in some places, to avoid being misleading. For example, the statement "In game-theoretic terms, deception should be considered (as it is, by the literature) to be a non-cooperative behaviour" would not seem to me to be always true. For

example, what about storytelling, surprise parties, and gift giving? These are not non-cooperative acts. Further, again the use of the word "should" here implies a normative position; is it not sufficient to say that it "is" considered this way, or that your work is concerned with the cases where deception is non-cooperative? Is there a deeper point you're trying to make, here?

2) Presentation of the Model and Experimental Design

Regarding the method chosen, this would seem to me to be an individual-based (i.e. agent-based) simulation to solve an evolutionary game theoretic model. However, despite the authors stating that it is an evolutionary game theoretic model, the setting is not described in the usual EGT sense (i.e. replicator dynamics describing changes to the proportions of traits in a population), but instead more a description of an ABM using discrete agents. Either would be fine for the purposes of this study, in my view, but some further clarity and precision in Section 3 would be welcome, in order to make the work more accessible to those coming from an EGT background.

With the methodology more explicit, a related question to this would be concerned with how the choice of methodology affects the sort of predictions that can be made. For example, in relation to the issues raised by Powers et al:

- Powers, S. T., Ekárt, A. & Lewis, P., 2018, Modelling enduring institutions: The complementarity of evolutionary and agent-based approaches, *Cognitive Systems Research*. 52, p. 67-81

Without some of the above clarifications, the work is still complete in terms of required details, but perhaps lacks some of the accessibility and insight it might otherwise provide.

Perhaps my main issue with the paper is that it presents an experimental study but without any explicit research questions or hypotheses. This makes it difficult to understand why the authors did the things they report on.

For example, when we get to the phrase "We model six different PGGs with different population compositions" I found myself just asking "why?", since this was not presented as part of an experimental design intended to answer any particular question. I think it would help to draw out hypotheses first, and then describe the experiments and how they are designed to test those hypotheses.

Similarly, when we get to the Deception Model, there are a number of psychological concepts (communicative skills, cognitive load, leakage of evidence) that are introduced and, while they may be interesting, they are not motivated by any particular hypothesis or stated research question in the paper. In general, I found myself asking "why?" several times through Section 4, as the authors described what they did.

Essentially, grounding the experimental work in a section earlier on, where research questions / hypotheses are laid out, would help to motivate and justify these design decisions. These are in fact present in the paper later on, in an implicit way, e.g.

- "We introduced Deceivers to check if they destabilise Cooperation"
 - "...in order to test if Deceivers emerge in systems where no punishment for deception exists"

but the reader has to wait quite a while to get this context. In general, the way the results section is written, particularly the assumptions parts, there are a number of implicit hypotheses that could be drawn out explicitly.

In terms of the results themselves, there are a number of interesting insights here, and these are drawn out well in the Discussion section. The exposition of the possible implications of these results in the context of issues of the day is also very welcome.

Minor issues:

- There appears to be some text missing at the end of the "Interrogator" bullet on page 5.
- The paper refers (on p5) to this being Cooperative Game Theory. Is that really the case? Note that there is a difference between studying the emergence of cooperation in a competitive game, and what is usually described as Cooperative Game Theory, the latter being typically concerned with the formation of coalitions. Some clarification here would help.
- Please ensure all figures have appropriate labels (e.g. chart axes). It would also be nice to read fuller captions explaining what insight each figure provides.
- One of the claims made is that "deception becomes the optimal strategy if communicative skill is maximised". What is meant by "optimal strategy" here? Is it a dominant strategy? A Nash Equilibrium strategy? An Evolutionary Stable Strategy? Again, use of established terms in (evolutionary) game theory would help the accessibility of the paper and its results.
- On page 6, the notion of trust is suddenly introduced. Trust is quite a loaded term, and has several different interpretations in various fields. I am not convinced by what is there at the moment that trust becomes relevant here, but if it is, then there needs to be a clearer exposition of what sort of trust you are talking about here (ideally taking a definition that applies to this work). As things stand, I found the passage that begins with "Trust has the following properties in our model..." rather confusing; it appeared with very little context.

Reviewer: 2

Comments to the Author(s)

The reviewer's comments' on the manuscript entitled
"The Evolution of Deception"

The topic of deception is very interesting in the fields of philosophy, psychology, economics and political science. This paper builds a new model combining Interpersonal Deception Theory and Truth-Default Theory, to analysis of the growth of deception in societies and the effectiveness of several approaches to reducing deception. The results of this paper indicate that cooperation in knowledge sharing can be re-established in systems by introducing institutions that investigate and regulate both defection and deception using a decentralized case-by-case strategy. The work done is interesting and the discussion seems fine. However, some issues should be addressed before the paper can be published.

Some comments are listed as follows:

- 1) The model part lacks the parameter range description and rationality analysis, that is, what are the value ranges of the parameters in each PGG? What are the dominant strategies for different value ranges? Are these settings reasonable?
- 2) The payoff functions of different strategies in every PGG are important channels for readers to understand the model proposed in this article. This should be shown in the manuscript rather than in the attachment.
- 3) The results of this paper are obtained through simulation, but there is no setting of initial conditions.
- 4) There is almost no result analysis. For example, in the results section of PGG2, the first sentence "Deceivers have indeed had an impact on the system" is followed by enumerating the data. However, how does deception affect the evolution of cooperation and how is it reflected in

the data? There is no explanation. Other results have similar problems, which will make it difficult for readers to understand.

5) What is the role of Loner? In the article DOI: 10.1126/science.1070582, the introduction of the loner can restart cooperation and avoid the dilemma of defection. So, since loner is introduced in this article, what is its effect on the results?

6) The "PeP" and "PoP" in Table 1 do not match their introductions, their order may be reversed, please check carefully.

7) References should be cited in order.

8) There are nine strategies in this paper, but you said "The eight strategies that we used..." on Page7 line29.

9) On Page 9 line22, the titles' formats of Table 2 and Table 1 are not uniform, one has a '.' at the end, while the other does not.

10) Table 2 is missing header row.

11) The location of the table and its introduction are too far away. The table should be placed in a suitable location for readers to view.

12) Similarly, the result pictures and their introduction are too far apart. In order to facilitate readers' reading, the pictures should be placed near the corresponding results. Besides, the captions of the pictures are posted too close, which affects the display of the picture.

13) What does "sboth" at page12 line5 mean?

===PREPARING YOUR MANUSCRIPT===

===PREPARING YOUR REVISION IN SCHOLARONE===

Author's Response to Decision Letter for (RSOS-201032.R0)

See Appendix B.

RSOS-201032.R1 (Revision)

Review form: Reviewer 1

Is the manuscript scientifically sound in its present form?

Yes

Are the interpretations and conclusions justified by the results?

Yes

Is the language acceptable?

Yes

Do you have any ethical concerns with this paper?

No

Have you any concerns about statistical analyses in this paper?

No

Recommendation?

Accept with minor revision (please list in comments)

Comments to the Author(s)

Thank you for the chance to review a revised version of your paper, and thanks also for your detailed response to my earlier comments. They were very helpful.

I am satisfied that the paper is publishable, and will represent an interesting and insightful read for many researchers.

I do have two small suggestions for final editing:

- It would be nice to mention the 'machine behavior' point earlier in the paper. The reference and mention of that agenda is only introduced in the conclusions, but is an implicit research context underpinning the whole work -- and its value in the current zeitgeist (as some of your footnotes quite rightly point out). Perhaps introduce that context in the introduction explicitly?

- I am still not convinced by the punishment part of this sentence: "Therefore mechanisms are required to promote cooperation in information dissemination and to punish misinformation and deception." in the Introduction. For sure, in this study, you have taken a game theoretic / PGG approach to modeling the evolution of deception, which relies on punishment to incentivize behavior. And that is of course necessary in order to make progress in a concrete model. Therefore, the discussion and exploration of punishment as a mechanism is appropriate in the bulk of the paper, as the specific mechanism studied. However, I am not convinced that the need for punishment as a mechanism in the PGG generalizes to this more broad statement about

requirements for punishment in society at large. There are several ways to bring about more cooperative behavior among groups of people, of which punishment is just one (and perhaps not always the most effective one). Others might include education, forgiveness, proactive inclusion interventions, empathy therapy... I do think we need to be careful not to tacitly impose the assumptions of a particular game theoretic model to humanity per se.

Thanks again for the chance to read about your work.

Review form: Reviewer 2

Is the manuscript scientifically sound in its present form?

Yes

Are the interpretations and conclusions justified by the results?

Yes

Is the language acceptable?

Yes

Do you have any ethical concerns with this paper?

No

Have you any concerns about statistical analyses in this paper?

No

Recommendation?

Accept as is

Comments to the Author(s)

From my viewpoint, this new version of the manuscript may be acceptable for publication in RSOS.

Review form: Reviewer 3

Is the manuscript scientifically sound in its present form?

No

Are the interpretations and conclusions justified by the results?

Yes

Is the language acceptable?

Yes

Do you have any ethical concerns with this paper?

No

Have you any concerns about statistical analyses in this paper?

Yes

Recommendation?

Reject

Comments to the Author(s)

It looks like I may be coming in late as a new reviewer after revisions were made.

I am approaching this review from the perspective of a researcher with a background in the social science of deception. I am not qualified to evaluate the modeling approach.

In the first paragraph under 2. Background, on page 3, the definition of deception as non-cooperative departs from some psychological and especially communication-centered definitions where deception can aid cooperation. Social agents, for example, are likely mindful other's face needs, smoothing over social disagreements can facilitate cooperation, and that blunt bald-faced honesty can be non-cooperative. Thus, it is probably wise to limit the discussion to the relevant subset of human deception that is exploitative. [I saw the footnote to reference 11, but this is not universally accepted].

For truth-default theory, it probably best to consult the most current and complete version (see Levine, 2020 "Duped").

Last paragraph under 2. Background, it seems odd to take and merge contradictory parts from different theories (TDT and IDT). Doing so runs afoul of both theories. I'm not sure this theoretical Frankenstein is the way to go.

Does it matter that "In the initial run of each simulation, the population starts with all agents being Defectors?" In truth-default theory, communication is honest by default and people on deceive when the truth means they cannot achieve their goals.

In lie prevalence research (Serota and colleagues; also TDT) most people are honest and most lies are told by a few prolific liars.

At the end of the day, the currently modeling does not seem to match what we know about the actual frequency of human-to-human deception.

Decision letter (RSOS-201032.R1)

Dear Dr Sarkadi

On behalf of the Editors, we are pleased to inform you that your Manuscript RSOS-201032.R1 "The Evolution of Deception" has been accepted for publication in Royal Society Open Science subject to minor revision in accordance with the referees' reports. Please find the referees' comments along with any feedback from the Editors below my signature.

We invite you to respond to the comments and revise your manuscript. Below the referees' and Editors' comments (where applicable) we provide additional requirements. Final acceptance of

your manuscript is dependent on these requirements being met. We provide guidance below to help you prepare your revision.

Please submit your revised manuscript and required files (see below) no later than 7 days from today's (ie 05-Aug-2021) date. Note: the ScholarOne system will 'lock' if submission of the revision is attempted 7 or more days after the deadline. If you do not think you will be able to meet this deadline please contact the editorial office immediately.

on behalf of Professor Bart de Moor (Associate Editor) and Marta Kwiatkowska (Subject Editor)
openscience@royalsociety.org

Reviewer comments to Author:
Reviewer: 1
Comments to the Author(s)

Thank you for the chance to review a revised version of your paper, and thanks also for your detailed response to my earlier comments. They were very helpful.

I am satisfied that the paper is publishable, and will represent an interesting and insightful read for many researchers.

I do have two small suggestions for final editing:

- It would be nice to mention the 'machine behavior' point earlier in the paper. The reference and mention of that agenda is only introduced in the conclusions, but is an implicit research context underpinning the whole work -- and its value in the current zeitgeist (as some of your footnotes quite rightly point out). Perhaps introduce that context in the introduction explicitly?

- I am still not convinced by the punishment part of this sentence: "Therefore mechanisms are required to promote cooperation in information dissemination and to punish misinformation and deception." in the Introduction. For sure, in this study, you have taken a game theoretic / PGG approach to modeling the evolution of deception, which relies on punishment to incentivize behavior. And that is of course necessary in order to make progress in a concrete model. Therefore, the discussion and exploration of punishment as a mechanism is appropriate in the bulk of the paper, as the specific mechanism studied. However, I am not convinced that the need for punishment as a mechanism in the PGG generalizes to this more broad statement about requirements for punishment in society at large. There are several ways to bring about more cooperative behavior among groups of people, of which punishment is just one (and perhaps not

always the most effective one). Others might include education, forgiveness, proactive inclusion interventions, empathy therapy... I do think we need to be careful not to tacitly impose the assumptions of a particular game theoretic model to humanity per se.

Thanks again for the chance to read about your work.

Reviewer: 2

Comments to the Author(s)

From my viewpoint, this new version of the manuscript may be acceptable for publication in RSOS.

Reviewer: 3

Comments to the Author(s)

It looks like I may be coming in late as a new reviewer after revisions were made.

I am approaching this review from the perspective of a researcher with a background in the social science of deception. I am not qualified to evaluate the modeling approach.

In the first paragraph under 2. Background, on page 3, the definition of deception as non-cooperative departs from some psychological and especially communication-centered definitions where deception can aid cooperation. Social agents, for example, are likely mindful other's face needs, smoothing over social disagreements can facilitate cooperation, and that blunt bald-faced honesty can be non-cooperative. Thus, it is probably wise to limit the discussion to the relevant subset of human deception that is exploitative. [I saw the footnote to reference 11, but this is not universally accepted].

For truth-default theory, it probably best to consult the most current and complete version (see Levine, 2020 "Duped").

Last paragraph under 2. Background, it seems odd to take and merge contradictory parts from different theories (TDT and IDT). Doing so runs afoul of both theories. I'm not sure this theoretical Frankenstein is the way to go.

Does it matter that "In the initial run of each simulation, the population starts with all agents being Defectors?" In truth-default theory, communication is honest by default and people on deceive when the truth means they cannot achieve their goals.

In lie prevalence research (Serota and colleagues; also TDT) most people are honest and most lies are told by a few prolific liars.

At the end of the day, the currently modeling does not seem to match what we know about the actual frequency of human-to-human deception.

===PREPARING YOUR MANUSCRIPT===

Your revised paper should include the changes requested by the referees and Editors of your manuscript. You should provide two versions of this manuscript and both versions must be provided in an editable format:
one version identifying all the changes that have been made (for instance, in coloured highlight, in bold text, or tracked changes);

===PREPARING YOUR REVISION IN SCHOLARONE===

- If you are requesting a discretionary waiver for the article processing charge, the waiver form must be included at this step.
- If you are providing image files for potential cover images, please upload these at this step, and inform the editorial office you have done so. You must hold the copyright to any image provided.
- A copy of your point-by-point response to referees and Editors. This will expedite the preparation of your proof.

- Ensure that your data access statement meets the requirements at <https://royalsociety.org/journals/authors/author-guidelines/#data>. You should ensure that you cite the dataset in your reference list. If you have deposited data etc in the Dryad repository, please only include the 'For publication' link at this stage. You should remove the 'For review' link.
- If you are requesting an article processing charge waiver, you must select the relevant waiver option (if requesting a discretionary waiver, the form should have been uploaded at Step 3 'File upload' above).
- If you have uploaded ESM files, please ensure you follow the guidance at <https://royalsociety.org/journals/authors/author-guidelines/#supplementary-material> to include a suitable title and informative caption. An example of appropriate titling and captioning may be found at https://figshare.com/articles/Table_S2_from_Is_there_a_trade-off_between_peak_performance_and_performance_breadth_across_temperatures_for_aerobic_scope_in_teleost_fishes_/3843624.

Author's Response to Decision Letter for (RSOS-201032.R1)

See Appendix C.

Decision letter (RSOS-201032.R2)

Dear Dr Sarkadi,

I am pleased to inform you that your manuscript entitled "The Evolution of Deception" is now accepted for publication in Royal Society Open Science.

on behalf of Professor Bart de Moor (Associate Editor) and Marta Kwiatkowska (Subject Editor)
openscience@royalsociety.org

Appendix A

King's College London
Department of Informatics
Bush House, WC2B 4BG
London, UK

June 5, 2020

To the Associate Editor,

We thank you for your comments to improve our manuscript '*The evolution of deception*', and for the chance to re-submit it to the journal Royal Society Open Science.

This paper addresses a the issue of deception and misinformation in agent societies. Understanding the interplay between deception and cooperation in knowledge sharing is critical if we, as a society, aim to mitigate its effects.

The method we used to perform the study is based on the evolutionary mechanism design of public goods games that also integrates solid theories of interpersonal deception from communication theory. Our results provide evidence for the adoption of methods for reducing the use of deception in the world around us.

To the best of our knowledge this is the first study of this kind. We believe that the implications of this work are far reaching and that the methods used would be of interest to a wide audience including those from artificial intelligence, systems theory, psychology, social learning, political science, communication theory, cognitive and behavioural sciences and others. Therefore we believe it would be a suitable candidate for publication in Royal Society Open Science.

Following the decision of reject and re-submit, we have substantially revised our manuscript considering the comments from Professor Bart De Moor, which are the following:

'Associate Editor Comments to Author (Professor Bart de Moor): The manuscript treats an important challenge that is highly relevant for today's society that is heavily impacted by social media, in which 'fake news' has an increasingly disturbing role.

While seemingly scientifically sound, the authors should do a better job in explaining how realistic and relevant all of the assumptions and simplifications are in the 6 PGG models presented, and how the results could possibly be implemented in realistic environments. While the introduction adequately refers to the current relevance of the research proposed, the conclusions in the section Results are highly quantitative, but the qualitative discussion in the Discussion part is rather terse, which complicates assessing the societal relevance and impact of the results. Also, it would be interesting to hear the authors opinion on how to implement the results in current mixed democratic-undemocratic socio-political systems on a global scale.

If - and this is stressed - the authors are able to address these concerns in a substantially re-

vised manuscript, we may be able to proceed further. Nevertheless, please accept our apologies for the unusual length of time taken to reach this recommendation - the handling Associate Editor has been involved in a lengthy period of travel.'

To do so, we have done the following:

1. We have introduced subsections named 'Model Assumptions' for each PGG in the Results section. In these subsections we have made explicit the assumptions that we make for each of the 6 PGGs.
2. We have re-written the 'Discussion' section such that we better reflect the qualitative results of our study and to ease the assessment of their societal relevance and impact. We have introduced Greco and Floridi's concept of *The Tragedy of The Digital Commons* (TDC) (Greco, 2004) in order to focus the discussion on knowledge as a public good, and to show how our results indicate a potential solution to avoid TDC. We have also introduced several real-world examples of platforms that represent potential implementations of PGG6 and discuss these potential implementations with regard to the concept of Habermas's *Public Sphere* (Habermas, 2015) and its role in the formation of public opinion.
3. We have added a 'Conclusions' section that is separate from the 'Discussion' section, and that summarises the paper.

Yours,

Stefan Sarkadi
Alex Rutherford
Peter McBurney
Simon Parsons
Iyad Rahwan

Bibliography:

Greco GM, Floridi L. *The tragedy of the digital commons*. Ethics and Information Technology. 2004 Jun 1;6(2):73-81.

Habermas J. *The theory of communicative action: Lifeworld and systems, a critique of functionalist reason*. John Wiley & Sons; 2015 Oct 7.

Appendix B

To the Associate Editor,

We thank you for your comments to improve our manuscript ‘The evolution of deception’, and for the chance to revise our submission according to the comments of the two reviewers.

We thank the reviewers for taking their time to read and assess the manuscript, and for offering comments of the greatest quality.

Following the decision, we have revised our manuscript considering the comments from the two reviewers to improve the paper.

To summarise, the main issues raised by the two reviewers relate to the presentation of the research questions and the results. These are: (i) the explicit formulation of research questions and hypotheses, and (ii) a clearer description of the experimental setup; both of which, if addressed, would improve the readability and interpretation of our study. To address these issues, we have done some general revisions of the paper.

The general revisions are the following:

1. We have introduced a new section named ‘Research Questions’. In this section we formulate the two major research questions RQ1 and RQ2 that drove our study. We also discuss other questions that can be derived and answered on the basis of the two major research questions.
2. We have moved the payoff’s functions from the Supplementary Material S1 into the main body of the paper as a subsection, at the end of the section named ‘Methods’. We named this subsection ‘Computing Payoff’s’.
3. We have introduced a subsection called ‘Experimental Setup’ inside the ‘Methods’ section, in which we describe the setups for all of the six PGGs. This subsection merges the first paragraph from the old ‘Results’ section with the description of the PGGs from the old ‘Methods’ section. We hope that this gives a clearer understanding of the experimental design.
4. We have introduced two hypotheses for each of the six PGG experiments in the ‘Results’ section. Each pair of hypotheses regards the long-run average frequency of cooperators and free-riders, where one hypotheses targets the weak/intermediate social learning and the other tackles strong social learning, respectively. The hypotheses pairs are discussed in the figure captions of each PGG experiment that shows the long-run average frequencies of the strategies from the 103 simulation runs.

Minor edits have been highlighted in yellow in the manuscript. Some of these are;

- Added missing text for the “Interrogator” bullet on pg.6.
- Fixed y-axis labels in the charts and ensured that figures have appropriate labels.

- Re-wrote the passage where Trust is introduced.
- Added header row to Table 2, now it is Table 1, as it is introduced earlier in the paper.
- Fixed additional typos.
- Edited the references such that they appear in the order of citation.

Attached below are our comment-by-comment responses to the two reviews.

Yours,
Stefan Sarkadi
Alex Rutherford
Peter McBurney
Simon Parsons
Iyad Rahwan

Review 1

Comments to the Author(s) This article aims at developing our understanding of how deception emerges in a society under competitive, i.e. evolutionary, pressures. It presents a model, based on the public goods game, in which a number of variations of deceptive behaviour are explored. The primary aim of the work is to attempt to identify ways of reducing deception. The topic of the article is an important and timely one, and its relevance is well contextualised in terms of the role of AI and social media in phenomena like fake news. The insights from the article have the potential to help us understand how to tackle issues that arise from this. The third paragraph of the introduction takes this a step further, and puts forward a perspective on how we ought to do this. This section, while I agreed with much of it personally, takes the step from a statement of facts towards a normative statement, for example stating that punishment is required. There are a few snuck premises in here, and a bit of a cleaner separation between the point of view offered about "what ought to be done" vs the context statements above, would be helpful. In general, there are two main areas I will address in my review, where I think the paper would benefit from further consideration.

Response

We thank you for your very detailed and helpful feedback! We are very pleased to hear that you have found the article timely and insightful. Regarding the normative statement, we have tried to address this in the introduction by cleaning up the distinction between it and the context statements. Overall, we greatly appreciated your review and we tried our best to address all of the points that you have raised in order to improve the paper.

Below we explain how we have addressed in the paper each of these points and we aim to clarify any ambiguities.

Positioning and Contextualisation of the Work

Considering the overall problem studied is one of knowledge management within the context of the evolution of cooperation, Section 2 might benefit from some further positioning of the authors' work in the context of that by other existing models of this. E.g., by Ober and collaborators:

- Burth Kurka D, Pitt J, Ober J, 2019, Knowledge management for selforganised resource allocation, ACM Transactions on Autonomous and Adaptive Systems, Vol:14

- Ober, J, 2008, Democracy and Knowledge. Princeton University Press.

Similarly, the work is clearly positioned in the domain of the establishment and sustainability of institutions for cooperation in multi-agent games (for example avoiding the tragedy of the commons), but there is little mention of the theory of institutions (e.g. Douglass North, Elinor Ostrom). Ostrom's work, in particular, has been operationalised in multi-agent systems, so would seem relevant. E.g.

- Pitt J, Schaumeier J, Artikis A, 2013, Axiomatisation of Socio-Economic Principles for Self-Organising Institutions: Concepts, Experiments and Challenges, ACM Transactions on Autonomous and Adaptive Systems

It would be useful for interested readers to be able to understand how these theoretical frameworks relate to your work.

While of course the work itself will necessarily consider deception in a way that is specific and restricted in scope (otherwise making progress would be very hard), the paper would benefit from some broader consideration of deception in some places, to avoid being misleading. For example, the statement "In game-theoretic terms, deception should be considered (as it is, by the literature) to be a non-cooperative behaviour" would not seem to me to be always true. For example, what about storytelling, surprise parties, and gift giving? These are not non-cooperative acts. Further, again the use of the word "should" here implies a normative position; is it not sufficient to say that it "is" considered this way, or that your work is concerned with the cases where deception is non-cooperative? Is there a deeper point you're trying to make, here?

R: Regarding knowledge management, we mention in the 'Introduction' that misinformation damages the openness and transparency needed for selfgovernance.

We do this by referring to - Burth Kurka D, Pitt J, Ober J, 2019, Knowledge management for self-organised resource allocation, ACM Transactions on Autonomous and Adaptive Systems, Vol:14.

Indeed, we have also added in the 'Introduction' a statement that our

approach is less concerned with finding the optimum way to organise a society and that it is more concerned with identifying the mechanisms that can ensure a functional society, similarly to the positions in

- Pitt J, Schaumeier J, Artikis A, 2013, Axiomatisation of Socio-Economic Principles for Self-Organising Institutions: Concepts, Experiments and Challenges, ACM Transactions on Autonomous and Adaptive Systems.

and in

- Herzberg, RQ. Elinor Ostrom's Governing the Commons: Institutional Diversity, Self-Governance, and Tragedy Diverted. The Independent Review. 2020 April 1; 24(4):627–636.

Presentation of the Model and Experimental Design

Regarding the method chosen, this would seem to me to be an individual-based (i.e. agent-based) simulation to solve an evolutionary game theoretic model. However, despite the authors stating that it is an evolutionary game theoretic model, the setting is not described in the usual EGT sense (i.e. replicator dynamics describing changes to the proportions of traits in a population), but instead more a description of an ABM using discrete agents. Either would be fine for the purposes of this study, in my view, but some further clarity and precision in Section 3 would be welcome, in order to make the work more accessible to those coming from an EGT background. With the methodology more explicit, a related question to this would be concerned with how the choice of methodology affects the sort of predictions that can be made. For example, in relation to the issues raised by Powers et al:

- Powers, S. T., Ek'art, A. & Lewis, P., 2018, Modelling enduring institutions: The complementarity of evolutionary and agent-based approaches, *Cognitive Systems Research*. 52, p. 67-81

Without some of the above clarifications, the work is still complete in terms of required details, but perhaps lacks some of the accessibility and insight it might otherwise provide.

Perhaps my main issue with the paper is that it presents an experimental study but without any explicit research questions or hypotheses. This makes it difficult to understand why the authors did the things they report on.

For example, when we get to the phrase "We model six different PGGs with different population compositions" I found myself just asking "why?", since this was not presented as part of an experimental design intended to answer any particular question. I think it would help to draw out hypotheses first, and then describe the experiments and how they are designed to test those hypotheses.

Similarly, when we get to the Deception Model, there are a number of psychological concepts (communicative skills, cognitive load, leakage of evidence) that are introduced and, while they may be interesting, they are not motivated by any particular hypothesis or stated research question in the paper. In general, I found myself asking "why?" several times through Section

4, as the authors described what they did.

Essentially, grounding the experimental work in a section earlier on, where research questions / hypotheses are laid out, would help to motivate and justify these design decisions. These are in fact present in the paper later on, in an implicit way, e.g.

- "We introduced Deceivers to check if they destabilise Cooperation"

- "...in order to test if Deceivers emerge in systems where no punishment for deception exists"

but the reader has to wait quite a while to get this context. In general, the way the results section is written, particularly the assumptions parts, there are a number of implicit hypotheses that could be drawn out explicitly.

In terms of the results themselves, there are a number of interesting insights here, and these are drawn out well in the Discussion section. The exposition of the possible implications of these results in the context of issues of the day is also very welcome.

R: Indeed the study is an agent-based simulation that solves an evolutionary game theoretic model. We hope to have clarified this in the paper.

We explain that the modelling of cognitive aspects that drive the behaviour of agents is the reason we chose to adopt what is called the content-based approach in - Powers, S. T., Ek'art, A. & Lewis, P., 2018, Modelling enduring institutions: The complementarity of evolutionary and agent-based approaches, Cognitive Systems Research. 52, p. 67-81. We hope this adds some clarification to the terms used to describe the methodology.

To address the main issue that you have raised, namely '*Perhaps my main issue with the paper is that it presents an experimental study but without any explicit research questions or hypotheses.*', we have first introduced an additional section named 'Research Questions' that explicitly raises two main research questions that we aim to answer in the paper, as well as research questions that can be derived from the two main ones and that we also aim to answer.

Second, we have described the experimental setup in a separate section before the 'Results'.

Subsequently, we have defined different pairs of research hypotheses for each of the six PGG experiments. The hypotheses in each pair target social learning, weak and strong, respectively.

We hope that the additional sections along with the pairs of hypotheses help the reader understand how the data from the simulations shows us when cooperation is established, broken, and re-established w.r.t, each PGG.

Minor issues

Q: - There appears to be some text missing at the end of the "Interrogator" bullet on page 5.

R: We have fixed this by adding the missing text.

Q: - The paper refers (on p5) to this being Cooperative Game Theory.

Is that really the case? Note that there is a difference between studying the emergence of cooperation in a competitive game, and what is usually described as Cooperative Game Theory, the latter being typically concerned with the formation of coalitions. Some clarification here would help.

R: We apologise for this misnomer. We have fixed this in the text. What we are concerned with is, as you mentioned, the study of the emergence of cooperation in a competitive game.

Q: - Please ensure all figures have appropriate labels (e.g. chart axes). It would also be nice to read fuller captions explaining what insight each figure provides.

R: We have ensured that all figures have appropriate chart axes, e.g., all the y-axes now show the 1.0 as the maximum frequency. We have also added fuller captions for the figures that show the long-run avg. frequencies of strategies for PGG1-PGG6.

Q: - One of the claims made is that "deception becomes the optimal strategy if communicative skill is maximised". What is meant by "optimal strategy" here? Is it a dominant strategy? A Nash Equilibrium strategy? An Evolutionary Stable Strategy? Again, use of established terms in (evolutionary) game theory would help the accessibility of the paper and its results.

R: Thank you for pointing this out. And yes, we meant that it is an evolutionary stable strategy (ESS). We have clarified this in the text.

Q: - On page 6, the notion of trust is suddenly introduced. Trust is quite a loaded term, and has several different interpretations in various fields. I am not convinced by what is there at the moment that trust becomes relevant here, but if it is, then there needs to be a clearer exposition of what sort of trust you are talking about here (ideally taking a definition that applies to this work). As things stand, I found the passage that begins with "Trust has the following properties in our model..." rather confusing; it appeared with very little context.

R: Thank you for pointing this out. We have re-phrased the respective paragraph and we have added some clarifications. We have defined trust to represent the 'truth-default state' of a society. According to Truth-Default Theory (TDT), individuals are in the truth-default state because they expect other individuals they interact with are cooperative and truthful.

Review 2

The topic of deception is very interesting in the fields of philosophy, psychology, economics and political science. This paper builds a new model combining Interpersonal Deception Theory and Truth-Default Theory, to analysis of the growth of deception in societies and the effectiveness of several approaches to reducing deception. The results of this paper indicate that cooperation in knowledge sharing can be re-established in systems by introducing institutions that investigate and regulate both defection and deception using a decentralized case-by-case strategy. The work done is interesting and the discussion seems fine. However, some issues should be addressed before the paper can be published.

Response

We thank you for taking your time to give us insightful and practical feedback! We are very pleased to hear that you have found the work and the discussion interesting. Below we explain how we have tried to address your comments in the paper.

Q: 1) The model part lacks the parameter range description and rationality analysis, that is, what are the value ranges of the parameters in each PGG? What are the dominant strategies for different value ranges? Are these settings reasonable?

R: We have added the parameter ranges in Table 1 (former Table 2).

We have taken some of the parameter values from PGGs from - Abdallah S, Sayed R, Rahwan I, LeVeck BL, Cebrian M, Rutherford A, Fowler JH. Corruption drives the emergence of civil society. J. R. Soc. Interface. 2014 Apr; 11(93):20131044.

The values for the new parameters that we introduce provide the most interesting results. That is why they have been selected.

Q: 2) The payoff functions of different strategies in every PGG are important channels for readers to understand the model proposed in this article. This should be shown in the manuscript rather than in the attachment.

R: Thank you for pointing this out. We have complied with your suggestion and we have moved the payoff functions in the main text of the paper in the subsection 'Computing Payoffs'.

Q: 3) The results of this paper are obtained through simulation, but there is no setting of initial conditions.

R: The initial conditions for the experiments were described in the first paragraph of the 'Results' section. We understand that this might have caused confusion. Thus, we have introduced a new section called 'Experimental

Setup' in which we describe the overall setup of the experiments and strategies setups for each PGG.

Q: 4) There is almost no result analysis. For example, in the results section of PGG2, the first sentence "Deceivers have indeed had an impact on the system" is followed by enumerating the data. However, how does deception affect the evolution of cooperation and how is it reflected in the data? There is no explanation. Other results have similar problems, which will make it difficult for readers to understand.

R: We hope to have clarified this by adding a new section 'Research Questions' along with adding pairs of explicit hypotheses for each experiment that guide the reader through our results and analysis.

Q: 5) What is the role of Loner? In the article DOI: 10.1126/science.1070582, the introduction of the loner can restart cooperation and avoid the dilemma of defection. So, since loner is introduced in this article, what is its effect on the results?

R: The role of the Loner is the same as in DOI: 10.1126/science.1070582. We have clarified this in the paper where we describe the Loner strategy.

Q: 6) The "PeP" and "PoP" in Table 1 do not match their introductions, their order may be reversed, please check carefully.

R: Thank you for pointing this out. We have fixed this now.

Q: 7) References should be cited in order.

R: We have fixed this. The references now appear in the order of citation.

Q: 8) There are nine strategies in this paper, but you said "The eight strategies that we used. . ." on Page7 line29.

R: Thank you for pointing this out. Yes, there are nine strategies. We have fixed this.

Q: 9) On Page 9 line22, the titles' formats of Table 2 and Table 1 are not uniform, one has a '.' at the end, while the other does not.

R: Thank you for pointing this out. We have fixed this.

Q: 10) Table 2 is missing header row.

R: Thank you for pointing this out. We have added a table row for Table 2, that is now Table 1.

Q: 11) The location of the table and its introduction are too far away. The table should be placed in a suitable location for readers to view.

R: Thank you for pointing this out. We now introduce Table 2 much earlier in the paper. By consequence it has now become Table 1.

Q: 12) Similarly, the result pictures and their introduction are too far apart. In order to facilitate readers' reading, the pictures should be placed near the corresponding results. Besides, the captions of the pictures are posted too close, which affects the display of the picture.

R: Thank you for pointing this out. We have tried as much as possible to introduce the pictures as close to their textual introductions as possible. We have fixed the distance between the captions of the pictures.

Q: 13) What does "sboth" at page12 line5 mean?

R: 'sboth' was a typo. We have fixed it.

Appendix C

Response to Reviewers

Reviewer: 1

Comments to the Author(s)

Thank you for the chance to review a revised version of your paper, and thanks also for your detailed response to my earlier comments. They were very helpful.

I am satisfied that the paper is publishable, and will represent an interesting and insightful read for many researchers.

I do have two small suggestions for final editing:

- It would be nice to mention the 'machine behavior' point earlier in the paper. The reference and mention of that agenda is only introduced in the conclusions, but is an implicit research context underpinning the whole work -- and its value in the current zeitgeist (as some of your footnotes quite rightly point out). Perhaps introduce that context in the introduction explicitly?

- I am still not convinced by the punishment part of this sentence: "Therefore mechanisms are required to promote cooperation in information dissemination and to punish misinformation and deception." in the Introduction. For sure, in this study, you have taken a game theoretic / PGG approach to modeling the evolution of deception, which relies on punishment to incentivize behavior. And that is of course necessary in order to make progress in a concrete model. Therefore, the discussion and exploration of punishment as a mechanism is appropriate in the bulk of the paper, as the specific mechanism studied. However, I am not convinced that the need for punishment as a mechanism in the PGG generalizes to this more broad statement about requirements for punishment in society at large. There are several ways to bring about more cooperative behavior among groups of people, of which punishment is just one (and perhaps not always the most effective one). Others might include education, forgiveness, proactive inclusion interventions, empathy therapy... I do think we need to be careful not to tacitly impose the assumptions of a particular game theoretic model to humanity per se.

Thanks again for the chance to read about your work.

Response 1:

Thank you for your rigorous and most insightful comments! We have greatly appreciated the quality of your feedback.

To address your final suggestions we have done the following:

- (i) Introduced machine behaviour in the Introduction section.
- (ii) Added an explicit note in the text of the introduction that aims to make the reader aware that punishment is not the only way to counter deception.

Reviewer: 2

Comments to the Author(s)

From my viewpoint, this new version of the manuscript may be acceptable for publication in RSOS.

Response 2:

Thank you for your decision and for the rigour of your feedback throughout the review process!

Reviewer: 3

Comments to the Author(s)

It looks like I may be coming in late as a new reviewer after revisions were made.

I am approaching this review from the perspective of a researcher with a background in the social science of deception. I am not qualified to evaluate the modeling approach.

In the first paragraph under 2. Background, on page 3, the definition of deception as non-cooperative departs from some psychological and especially communication-centered definitions where deception can aid cooperation. Social agents, for example, are likely mindful other's face needs, smoothing over social disagreements can facilitate cooperation, and that blunt bald-faced honesty can be non-cooperative. Thus, it is probably wise to limit the discussion to the relevant subset of human deception that is exploitative. [I saw the footnote to reference 11, but this is not universally accepted].

For truth-default theory, it probably best to consult the most current and complete version (see Levine, 2020 "Duped").

Last paragraph under 2. Background, it seems odd to take and merge contradictory parts from different theories (TDT and IDT). Doing so runs afoul of both theories. I'm not sure this theoretical Frankenstein is the way to go.

Does it matter that "In the initial run of each simulation, the population starts with all agents being Defectors?" In truth-default theory, communication is honest by default and people do not deceive when the truth means they cannot achieve their goals.

In lie prevalence research (Serota and colleagues; also TDT) most people are honest and most lies are told by a few prolific liars.

At the end of the day, the currently modeling does not seem to match what we know about the actual frequency of human-to-human deception.

Response 3:

We thank you for your general feedback coming from the social sciences.

To address your comment about deception being non-cooperative, we have specified in the introduction of the paper that we mostly wished to target the negative effects of deception on the knowledge as a public good, e.g. what you call exploitative deception. We also agree that deception in social contexts can also play a positive role by smoothing cooperation, but that is beyond the scope of our paper.

We have also added the most recent reference on TDT, namely Levine's book entitled "Duped", see reference [3].

Regarding the integration of the two theories, IDT and TDT, we believe that it is worthwhile to explore approaches that integrate their components.

The reason we have started with all agents being Defectors is to make our work an extension of existing work in the literature, specifically [27] and [28]. In that work, the simulations of PGGs start from all Defectors. We wished to reproduce these results and show how cooperation can be re-established. Additionally, note that Defectors do not deceive or attempt deception at all, they just defect in the context of the PGG of [27, 28]. In that sense they are quite open about their free-riding in the PGG.

It is important to note that we have modelled trust according to TDT. In our model trust is affected by the prevalence of Cooperators, namely the more agents cooperate, the higher the degree of trust. Conversely, the more Deceivers or Defectors there are, the lower the degree of trust. This way of modelling trust is in accordance with TDT.

Regarding lying prevalence research, we thank you for the suggested reference by Serota on TDT which we have added, see reference [48]. We believe that it might be very interesting to pursue future work that explores the relation between our model and empirical research on deception prevalence in humans. For example, the calibration of our model based on the insights by [49] in order to be able to represent deception and to match the data in human-to-human interactions. Currently our model is tailored for explaining deception in hybrid societies where machines and humans interact.